# Nafion: New and Old Insights into Structure and Function

**DOI:** 10.3390/polym15092214

**Published:** 2023-05-07

**Authors:** Barry W. Ninham, Matthew J. Battye, Polina N. Bolotskova, Rostislav Yu. Gerasimov, Valery A. Kozlov, Nikolai F. Bunkin

**Affiliations:** 1Department of Materials Physics, Research School of Physics, Australian National University, Canberra, ACT 2600, Australia; 2Breakthrough Technologies, Deakin, ACT 2600, Australia; 3Department of Fundamental Sciences, Bauman Moscow State Technical University, 2-nd Baumanskaya Str. 5, Moscow 105005, Russia

**Keywords:** fuel cells, photoluminescence spectroscopy, Fourier transform IR spectroscopy, swelling of polymer membrane, Nafion, unwinding of polymer fibers, exclusion zone, endothelial surface layer, deuterium-depleted water, specific electrolyte (Hofmeister) effects

## Abstract

The work reports a number of results on the dynamics of swelling and inferred nanostructure of the ion-exchange polymer membrane Nafion in different aqueous solutions. The techniques used were photoluminescent and Fourier transform IR (FTIR) spectroscopy. The centers of photoluminescence were identified as the sulfonic groups localized at the ends of the perfluorovinyl ether (Teflon) groups that form the backbone of Nafion. Changes in deuterium content of water induced unexpected results revealed in the process of polymer swelling. In these experiments, deionized (DI) water (deuterium content 157 ppm) and deuterium depleted water (DDW) with deuterium content 3 PPM, were investigated. The strong hydration of sulfonic groups involves a competition between ortho- and para-magnetic forms of a water molecule. Deuterium, as it seems, adsorbs competitively on the sulfonic groups and thus can change the geometry of the sulfate bonds. With photoluminescent spectroscopy experiments, this is reflected in the unwinding of the polymer fibers into the bulk of the adjoining water on swelling. The unwound fibers do not tear off from the polymer substrate. They form a vastly extended “brush” type structure normal to the membrane surface. This may have implications for specificity of ion transport in biology, where the ubiquitous glycocalyx of cells and tissues invariably involves highly sulfated polymers such asheparan and chondroitin sulfate.

## 1. The Problem

The perfluorinated polymer membrane Nafion™ is used in fuel cells for hydrogen energy. The membrane has very high thermal, mechanical and chemical stability [1]. This is used not only in hydrogen energetics, but also in physics and chemistry, see, e.g., [2,3,4,5,6,7,8,9,10]. In contact with aqueous solutions Nafion has peculiar bulk and surface ultrastructural properties that appear utterly unlike anything else. It remains mysterious. Nafion is produced by copolymerization of a perfluorinated vinyl ether co-monomer with tetrafluoroethylene (Teflon). Below we present the chemical formula of Nafion [1]:



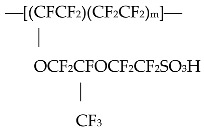



The Teflon moiety is essentially hydrophobic, while the sulfonic SO_3_H groups exhibit hydrophilic properties. Connected water-filled helical channels with diameter of 2–3 nm form within the Nafion matrix during swelling in water [1]. The interior aqueous nanocores conduct cations. The channels are negatively charged and strongly hydrated due to the dissociation of terminal sulfonic groups via R—SO_3_H + H_2_O ⇔ R—SO_3_^−^ + H_3_O^+^. The protons reside inside these water-filled channels due to the presence of negative charge, distributed over their interior surface. Such a nanostructure is the key to understanding the spatial separation of H^+^ and OH^−^ ions in low-temperature hydrogen power plants [10]. It is important that the Teflon base provides excellent mechanical and chemical stability of the membrane, while the mobile protons resulted from the dissociation of the sulfonic acid groups, provide active sites for cation conduction [11,12,13].

Here are some characteristics of prime interest. The nanostructure of the polymer matrix that occurs on swelling, consists of conducting cylindrical channels reminiscent of reverse phase microemulsions. As shown in Refs. [14,15,16,17,18], the microstructure of bicontinuousmicroemulsions consists of water tubes covered by surfactant molecules in oil. These microemulsions are formed from double chained ionic surfactants, water and alkanes. The water nanotubes contain counterions and have multiple connections. Characteristic patterns of such structures are given in Figure 1 and Figure 2. In Figure 1 we show cryo-SEM micrograph of an iso-octane/DDAB surfactant/water microemulsion(reprinted from [16] with permission), and in Figure 2 we display the structure of micro emulsion formed from double chained cationic surfactant didodecyl dimethyl ammonium bromide (DDAB) water and alkane (adapted from [17,18] with permission). Depending on component ratios, their diameter can vary between 2–10 nm. Conductivity and viscosity can vary over 10 and 5 orders of magnitude, respectively. With different proportions of the components different kinds of nanostructures structures can spontaneously form. Two and three phases can be in equilibrium and can transit from one form to another. This variability in the self-assembled structure is driven by two effective forces. These are local curvature at the water–oil interface which set by inter-surfactant molecular forces, and global packing constraints.

### 1.1. The Nafion Exclusion Zone and the 4th State of Water Controversy

Most research has focused on the bulk properties of Nafion after it has been swollen in water. What happens at the surface is even more interesting. As was shown in Ref. [19], a special area forms in the bulk water adjacent to the Nafion surface. This area is calledthe exclusion zone (EZ). The EZ extends for hundreds of microns. It is very dilute. It expels micron-sized colloidal particles and red blood cells. The EZ structure does not change over several days. The EZ size in heavy water (D_2_O) is slightly less than in DI water [20]. Its existence is inexplicable by classical theories of physical chemistry.

A first attempt [21] to demystify the physical nature of EZ was an experiment, in which the absorption of a light wave directed in parallel to the surface of Nafion in grazing incidence geometry was investigated. The beam of a laser diode with a wavelength of 270 nm directed parallel to the Nafion surface in water, was shifted towards the surface. The absorptivity of water adjacent to the Nafion surface increased as the laser beam approaches the surface. The effect has been attributed, controversially, to the absorption of light at wavelength λ = 270 nm by water molecules, which were supposed to exist in a new (4th) state of water in the EZ. This can be ruled out. It has been shown in [22] that water molecules do not have absorption lines near 270 nm. Further, in one of the first studies of the absorption spectra of Nafion in the visible and near UV ranges, it was found that both dry Nafion and Nafion soaked in water absorb radiation at wavelength λ = 270 nm, see Ref. [23]. It then seems reasonable to associate the absorption of radiation within the EZ with the presence of Nafion particles. The proposed anomalous water is unnecessary. The question of the origin of Nafion particles within a region separated from the Nafion surface at a distance of the order of hundreds of microns will be studied in detail below.

What was envisaged in [19,20,21], was that the Nafion surface imposes a new quasi-crystalline structure on the adjacent layers of water, and this structure was supposed to exist on a macroscopic scale. This “fourth phase” of water represents a revival of Derjaguin’spolywater. If true, this would be very important. It is reminiscent of the well-known and still unexplained works [24,25,26,27], devoted to stable suspensions of heavily dialyzed latex spheres. As shown in these articles, the distance between these spheres amounts to hundreds of nanometers. The EZ effects with Nafion [19,20,21] are real. In what follows, a quantitative interpretation of the effects associated with the EZ will be proposed. Our work shows that the EZ has an even more exotic, but, we believe, a more credible origin.

### 1.2. The Deuterium Effect and Chirality

To explore further the structure of water close to the polymer surface, we used a novel instrument for laser diagnostics developed earlier. It is based on photoluminescence spectroscopy, and is augmented by fiber optics, see [28].

This technique was used to study a Nafion N117 membrane with thickness of 175 μm at room temperature (22 °C). It was swollen in water with a deuterium content varying from 3 to 10^6^ ppm. This range includes all possible concentrations of deuterium in water. In DI water, the deuterium content is 157 ± 1 ppm, which is the average standard for oceanic water (SMOW), see [29].

The use of deuterium as a probe of microstructure had an ancillary motivation. Yosef Scolnik in his very interesting paper [30] suggested, and gave good reason to suppose, that the sources of chirality could be traced to the ratio of the two kinds of molecular states in water: para-magnetic or ortho-magnetic forms. They can be expected to bind to solute molecules competitively and cooperatively and to impose a handedness that produces chirality. Note that in the equilibrium vapor phase the ratio between the content of ortho-magnetic and para-magnetic spin isomers is exactly equal to 3:1, see, for example, [31].

The winding up of Nafion fibers in water in a helical form, required to produce nanochannels in the bulk of polymer, would depend on that ratio. It seems that deuterium somehow displaces the bound water in the polymer matrix, and so could be useful in probing nanostructure.

In our previous work [32], the isotopic effects, revealed upon swelling Nafion in water with various content of deuterium, have been explored in experiments with photoluminescent spectroscopy. The experimental schematic for photoluminescent spectroscopy, described here, is quite similar to the original setup, used in Ref. [32]. In addition, the dynamics of Nafion swelling in water was studied by us in Fourier Transform IR (FTIR) spectroscopy experiments, see [33], i.e., we studied the dynamics of Nafion swelling in liquids with different deuterium contents using spectroscopic techniques in both the IR and visible ranges.

### 1.3. A Parallel Universe; the Biological Analogue between the Exclusion Zone and the Endothelial Surface Layer, Nafion and the Glycocalyx

The peculiar properties of Nafion have an uncanny resemblance to those of the ubiquitous glycocalyx and endothelial surface layer of biology, see [34,35].

The glycocalyx (GC) covers the surface of veins and arteries of all tissues of the body, including presumably neurons. It adjoins the lipid membrane of all cells. The GC of the veins repels T-cells, cancer cells and lipoproteins. It is typically about 50 nm thick, less for red blood cells at 20 nm. (The phospholipid bilayer membrane of cells is only about 3-nm thick). Recall that Nafion is formed from a sulfonated fluorocarbon CF_2_- polymer, Teflon. The glycocalyx is mostly a mixture of two hydrocarbon sialic acid-based polymers, chondroitin sulfate and heparan sulfate, admixed in lesser amounts with hyaluronic acid. A key common factor is the extensive presence of sulfate and sulfonic groups in both.

The EZ for Nafion is enormous, extending up to 300 microns. The glycocalyx also has an exclusion zone called the endothelial surface layer (ESL) that extends about a micron. Like the EZ, it repels red blood cells, bacteria, cancer cells and colloidal particles. Its structure and function were completely unknown until recently [34]. Like the EZ of Nafion, it has sparce threads of polymer stretching from the GC surface. The open space between the threads is made up mainly of nanobubbles formed by passage of molecular carbon dioxide produced by metabolism through the polymeric frit provided by the GC. The CO_2_ nanobubbles are 100% toxic to viruses such as COVID-19 and to bacteria. They are stabilized at physiological salt concentrations. The ESL presumably forms the blood brain barrier, see Refs. [34,35] for more detail.

The glycocalyx conducts ions and hydronium just like Nafion and may even provide the basis of a complementary neural network. It also evidently forms a conducting connected network for ions such as Na^+^, K^+^, Ca^++^ and H_3_O^+^. (It is not known yet if the very hydrophobic strands of the EZ of Nafion also comprise nanometer-sized or micrometer-sized bubbles of gas). These observations are entirely new. It seems likely that work on properties of Nafion which is, if we like, the glycocalyx, “painted on a larger canvas”, will be useful in physiology.

## 2. Deuterium Effects Again: Materials and Methods

### 2.1. Reagents

In the experiments that we come to now, we use samples of deuterium depleted water (DDW; deuterium content ≤3 ppm), purchased from Sigma-Aldrich, St. Louis, MO, USA. Moreover, we explored the samples of deionized (DI) Milli-Q water (specific resistance 18 MΩ·cm at 25 °C), and samples of D_2_O, purchased from Cambridge Isotope laboratories, UK. For these samples, the deuterium content is 99.9999%, i.e., ~10^6^ ppm. Aqueous samples with different deuterium contents were prepared with the help of proportional volumetric mixing of D_2_O and DDW. The Nafion N117 plates (DuPont, Wilmington, DE, USA, plate thickness 175 μm) were soaked in these liquid samples.

We also investigated aqueous solutions of the heparin sulfate [C_12_H_19_NO_20_S_3_]_n_, having weight content 58 mg/mL (purchased from Sigma-Aldrich, St. Louis, MO, USA) and chondroitin sulfate [C_14_H_21_NO_15_S]_n_, having weight content 100 mg/mL (purchased from Belmedpreparaty, Minsk, Belarus). The idea of experiments with these substances is the presence of sulfonic groups SO_3_H, see below. Additionally, we explored reagent-grade salts LiCl, NaCl, KCl and CsCl, purchased from Sigma Aldrich, St. Louis, MO, USA. All these solutions were studied as prepared.

### 2.2. Optimal Wavelength Selection for Optical Pumping (Technical)

The absorption spectra of dry Nafion were measured using a GBC Cintra 4040 Spectrophotometer (UVISON Technologies Limited, Kent, UK); the spectral range is 190–900 nm. In this device, the slit width can be changed in the range from 0.1 to 2.0 mm, and the presence of a double Littrow monochromator in the Czerny–Turner configuration guarantees very high sensitivity, low stray light and baseline drift. The measured absorption spectra, corrected for Rayleigh scattering (such a technique of correction is described in detail, for example, in Ref. [36]), display two absorption bands centered at λ_1_ = 232 and λ_2_ = 268 nm, see Figure 3. The absorptivity is scaled in arbitrary units (a.u. hereinafter).

Luminescence can be excited provided that the pump radiation falls within one of the absorption bands of the substance under study, see Ref. [37]. In addition, the pump radiation must not be absorbed in water; otherwise, the radiation will be significantly attenuated in the liquid and, in addition, the absorption of the pump radiation will result in uncontrolled local heating effects and the associated convective counter-flow of liquid. As shown in [22], absorption in water in the wavelength range 196 to 320 nm can be neglected. We do not have a source of light at a wavelength that would fall inside the absorption band λ_1_ = 232 nm. To excite luminescence inside the absorption band centered at the wavelength λ_2_ = 268 nm, we used the fourth harmonic of radiation of a pulsed-periodic YAG:Nd^3+^ laser (“Laser Compact”, Russia, model DTL-382QT) with wavelength λ = 266 nm with the pulse repetition rate 3 kHz and the pulse duration 5 ns. The average pulse energy was 4 μJ. In addition, we used a continuous wave (CW) laser diode at a wavelength of λ = 369 nm with an average power of 5 mW. The radiation at this wavelength belongs to the long-wavelength wing of the absorption band centered at the wavelength λ_2_.

In Figure 4 we exhibit the luminescence spectrum of dry Nafion, excited with the pump at wavelengths λ = 266 and 369 nm. The exposure time for each measurement of spectrum was 8 s. As is seen, the obtained graphs are quite identical.

As is seen in this figure, the spectral maximum in both graphs is related to λ = 508 nm. Therefore, in what follows we will study the behavior of the luminescence intensity precisely at this wavelength. It turned out that when a polymer membrane is irradiated with pump radiation at a wavelength λ = 266 nm, degradation of the polymer from heating due to absorption occurs (this wavelength falls into the center of the absorption band λ_2_ = 268 nm). At the same time, when the membrane is irradiated at a wavelength of λ = 369 nm (the long-wavelength edge of the λ_2_ band), thermal degradation of the polymer does not happen. Therefore, in what follows, we will use pump radiation at this wavelength. This choice is due, among other things, to the fact that the spectral minimum of the absorptivity in water is related to λ = 370 nm, see, for example, Ref. [22].

### 2.3. Aspects and Elucidation of the Physical Nature of Luminescence (Technical)

As shown above, the Nafion consists of a tetrafluoroethylene (Teflon) backbone, terminated with sulfonic groups SO_3_H. Preliminary spectroscopic observations showed that Teflon, irradiated at λ = 369 nm, does not emit a luminesce signal in the spectral range close to this wavelength. Thus it is reasonable to assume that the luminescence centers in the case of Nafion are precisely the sulfonic groups.

To verify this hypothesis, we measured the luminescence signal from aqueous solutions of heparin sulfate and chondroitin sulfate, as well as from a solution of Nafion in isopropanol. The idea of investigating heparin and chondroitin sulfate is that these substances contain sulfonic group SO_3_H. The additional reason of why we investigated the luminescence spectra of aqueous solutions of heparin and chondroitin sulfate is that heparin and chondroitin sulfate are well known drugs. Since we studied the properties of Nafion in the context of possible biological applications, it was interesting for us to compare the luminescence spectra of Nafion, heparin and chondroitin sulfate.

We also studied the luminescence spectra from the solution of Nafion in isopropanol, and from the surface of a Teflon plate immersed in water upon irradiation at wavelength λ = 369 nm. The motivation for experiments with the solution of Nafion in isopropanol consisted in that Nafion can be dissolved in this liquid. Thus, we can change the concentration of Nafion in the “Nafion–isopropanol” mixture, which allow us to explore the luminescence intensity from this mixture vs. the content of dissolved Nafion.

To prepare Nafion solutions in isopropanol, a Nafion plate with an area of 6 cm^2^ and a thickness of 175 μm was soaked in isopropanol for 22 h. This plate was partially dissolved; the residue of the plate was removed from the liquid after 22 h of dissolution.

The spectra of luminescence from heparin, chondroitin sulfate, Teflon and the solution of Nafion in isopropanol upon irradiation at λ = 369 nm are shown in Figure 5. The profiles of spectra for Nafion, heparin sulfate and chondroitin sulfate are qualitatively similar. This similarity and the absence of the luminescence from Teflon is an explicit confirmation that the sulfonic groups SO_3_H serve as luminescence centers in all three substances.

Below, we investigate solutions of Nafion in isopropanol with various concentrations. For each solution, the experimental points and the calculated confidence intervals were determined by averaging of 10 measurements. In Figure 6 we present the dependence of the luminescence signal *P* from the Nafion solution in isopropanol, induced by the optical pump at a wavelength of λ = 508 nm, which is related to the spectral maximum of the luminescence, see Figure 4. The origin of the abscissa axis corresponds to pure isopropanol. The initial concentration of the Nafion solution was not known; we conditionally put that immediately after removal of the Nafion plate from the solution the concentration is equal to 100 arbitrary units (a.u).

The rectilinear segment (Figure 6) can be approximated as
*P* = −237 + 16*ρ*,(1)
where *ρ* is the volume number density of the luminescence centers. These centers are sulfonic groups, in accordance with our model. Since these groups are attached to polymeric fibers, *ρ* can be associated with the volume number density of Nafion particles in isopropanol solution, which corresponds to the abscissa axis in Figure 6. The linear relationship between the luminescence intensity *P* and the volume number density of sulfonic groups is very important. The luminescence intensity *P* can be represented as
(2)$$P=A+kI_{pump}\sigma_{lum}\rho V,$$
where *I_pump_* is the pump intensity, *A* = 20–270 relative units (r.u.) corresponds to the spectral density of the mini-spectrometer noise and stray-light intensity, *k* is the dimension factor, which characterizes the spectrometer, *V* is the luminescence volume and *σ_lum_* is the luminescence cross section. The straight line dependence in Figure 6 can be implemented provided that *σ_lum_* = const. The dimensional constants included in Equation (2) are taken into account by the factor *k*:(3)$$\;k=\left\{ \begin{array}{l} k_{0}=const,W>W_{thr}\\ 0,W\leq W_{thr} \end{array}\right.$$

Here *W_thr_* is the minimum intensity of the luminescence (the threshold value) that can be detected by our spectrometer, *W_thr_* ≡ *I_pump_σ_lum_ρV*.

Since the luminescence cross section *σ_lum_* is constant, the volume *V* is constant as well (this is a cylinder, the radius of which is equal to the radius of optical pump beam, and its height is equal to the height of the Nafion plate), and the liquid molecules penetrate into the near-surface layer of the polymer membrane upon swelling, the density *ρ* of the Nafion particles inside the volume *V* should decrease. Thus we can deduce the equation:(4)$$\frac{d\rho }{dt}=-\frac{\rho }{\tau },$$
where *τ* is the characteristic time for penetration of the liquid molecules inside the polymer. Thus we arrive at
(5)$$\rho \left(t\right)={\left(\rho \right)}_{0}\exp\left(-\frac{t}{\tau }\right).$$

Since *I*(*t*) ∝*ρ*(*t*), the luminescence intensity *I*(*t*) in the case of *σ_lum_*= *const* obeys an exponential decay law. 

### 2.4. Influence of Deuterium on Nafion Swelling

As will be shown below, the luminescence intensity in the spectral maximum depends on the deuterium content. This could occur due to isotopic quenching of luminescence, which results in decreasing the cross-section *σ_lum_*. To estimate the possible role of isotopic quenching, we performed the following experiments. The solutions of Nafion in isopropyl alcohol were diluted with water with a deuterium content 3 ppm (DDW), 157 ppm (DI water) and 10^6^ ppm (heavy water, D_2_O). The corresponding intensity dependences are shown in Figure 7. For convenience of presentation, the experimental errors are indicated only for DDW dependence, since these errors are approximately the same for each test liquid. The volume content of isopropyl alcohol is plotted along the abscissa. The zero abscissa corresponds to pure water. It is seen that the behavior of the luminescence intensity is the same for all liquid samples within experimental error. So the isotopic effects of luminescence quenching can be ignored.

## 3. Laser Luminescence Diagnostics of Nafion in Liquids (Technical)

The fact that the luminescence centers are terminal sulfonic groups located along the fluorocarbon polymer fibers allows us to make an optimal choice for the pumping geometry during photoluminescence excitation. As was shown in Ref. [38], the Nafion fibers, localized at the Nafion–water interface, are oriented predominantly normal to this interface. Since sulfonic groups (luminescence centers) are localized at the ends of polymer fibers (see [1]), it seems reasonable to excite photoluminescence from the Nafion surface in the geometry of grazing incidence of the optical pump beam, since in this case the largest number of luminescence centers is captured by the pump.

The experiments with the photoluminescence from Nafion were carried out via two protocols.

### First Protocol of Photoluminescence Experiments

A Nafion plate was placed in an empty cell in such a way that the pump radiation (optical axis) was directed along the Nafion surface in a grazing incidence configuration. Then liquid was poured into the cell. The moment of pouring corresponds with the beginning of the time *t* countdown. The luminescence signal *P* was measured as a function of soaking time *t*. This protocol included soaking Nafion in water with different deuterium contents for 30 min. After 30 min soaking, the Nafion plate was moved away from the optical axis with a step of 25 µm; the required number of steps was determined by the decrease in luminescence to zero. Thus, in this experiment, the luminescence signal was studied as a function of the distance *X* between the Nafion surface and the optical axis; the time of a single measurement was of the order of several seconds.

The schematic of the experimental setup for the first protocol has been described in detail Ref. [32] and is shown in Figure 8.

The probing radiation of the continuous wave laser diode (1) (optical pumping) at a wavelength λ= 369 nm was input to the multimode quartz optical fiber (2), having a diameter ∅ = 50 μm and a numerical aperture NA = *m*_0_⋅sinα = 0.3. Here *m*_0_ = 1 is the refractive index of air, α is the angle of beam divergence at the exit edge of the fiber in air. The fiber was fixed in a hole located in the center of the bottom of a cylindrical cell (3) made of Teflon; the direction of the pump beam established the optical axis in the cell. The cell was thermo-stabilized at room temperature (*T* = 23 °C) with the help of liquid thermostat (accurate to ±0.1 °C) and filled with liquid sample. A square-shaped Nafion plate (4) with a side *h* = 4 mm and a thickness *d* = 175 μm was placed inside the cell (3). The Nafion plate was rigidly fixed in parallel with respect to the optical axis, which provided the geometry of grazing incidence of the experiment. Upon immersion in water, the initially hydrophobic Nafion plate bends across the vertical axis due to the interfacial tension forces. However, this bending led only to an effective shift of the Nafion-water boundary (such a shift amounted to about 1 mm), but did not result in changing the angle of incidence of the pump radiation. The luminescence radiation reflected from the internal faces of the cell (Nafion is transparent in the visible range), collected along the optical axis of the cell, and was captured by multimode quartz fiber (5) fixed at the center of the cell the same way as the fiber (4). The fiber (5) transferred the luminescence signal to FSD-8 mini-spectrometer (6). To avoid too intense laser radiation from entering the mini-spectrometer input, the fiber (5) was equipped with a refocuser and a light filter that cut-off radiation within the spectral range λ < 350 nm (not shown in Figure 8). The duration of a single measurement was 7–8 s. The experimental data were accumulated by a computer (7).

In this experiment, we studied the spatial profile of the luminescence intensity in its spectral maximum (508 nm) from the surface of dry Nafion, as well as from the surface of Nafion, which was soaked in the test liquid for 30 min. In both cases, the Nafion plate was displaced normal to the optical axis by means of a stepper motor (8). Each step corresponded to a shift of 25 μm. The necessary number of steps was determined from the requirement of decay of the luminescence signal to zero level, i.e., luminescence intensity was investigated as a function of the distance *x* between the Nafion surface and the optical axis. The reference point *x* = 0 was determined from the condition of maximum luminescence signal; in this case the optical pump beam moves along the Nafion surface.

## 4. Theory for the Luminescence Measurement Technique (Technical)

In the experiments carried out in accordance with this protocol it was necessary to take into account the spatial profile of the pump intensity *I_pump_*(*x*) and the spatial distribution of the volume number density of Nafion particles *ρ*(*x*). In this connection Equation (2) must be rewritten in the form
(6)$$P\left(x\right)=A+\int_{-\infty }^{+\infty }G\left(x-x_{1}\right)\cdot \rho \left(x_{1}\right)dx_{1}.$$

Here the symmetric kernel *G*(*x* − *x*_1_) of this integral equation stands for the apparatus function for the experimental setup, and all dimensional factors, included in Equation (2), are automatically accounted by the kernel *G*(*x* − *x*_1_). Our goal is to obtain an explicit expression for *G*(*x* − *x*_1_), which will then allow us to deduce the spatial distributions *ρ*(*x*) of the Nafion particles in the bulk of liquid after the soaking of Nafion in water with different deuterium contents.

An example of a solution to Equation (6) for dry Nafion is shown in Figure 9; the value *x* = 0 corresponds to the boundary of dry Nafion. In this case, the distribution *ρ*(*x*) can be approximated by two Heaviside functions *θ*(*x*) as
(7)$$\rho (x)=\rho_{0}\cdot \left[\theta (x+d)-\theta (x)\right],$$
where *d* = 175 μm is the thickness of the Nafion plate, and *ρ*_0_ is a dimensional constant. Assuming that the pump radiation at the fiber output is divergent, and the profile *I_pump_*(*x*) is described by a Gaussian function, we take $G(x)=G_{0}\exp\left[-\frac{x^{2}}{2a^{2}}\right]$, where *G*_0_ is some dimensional constant, and the parameter *a* is the width of Gaussian profile, which we should find. 

After integration in Equation (6) we arrive at
(8)$$P(x)=A+G_{0}n_{0}\frac{a\sqrt{\pi }}{2}\left[\mathrm{erf}\left(\frac{x+d}{\sqrt{2a^{2}}}\right)-\mathrm{erf}\left(\frac{x}{\sqrt{2a^{2}}}\right)\right]$$

Here *A* = 124 a.u. is the average level of the apparatus noise (stray light), and the second term represents the integral convolution of Gaussian and Heaviside functions. Minimizing the discrepancy functional between the theoretical curve *P*(*x*) and the experimental points (see Figure 9), we obtain for the width of the apparatus Gaussian function *a* = 84 μm. Hereafter, *ρ*(*x*) is normalized to initial square of the Nafion plate, and therefore is scaled in normalized units, n.u.

### 4.1. Influence of Deuterium in Growth of the Exclusion Zone

When Nafion is soaked in water, the divergence of the pump beam decreases as compared with the divergence in air. Indeed, the divergence in air is described by the formula NA = *m*_0_⋅sinα, where NA = 0.3 is the numerical aperture of the optical fiber, *m*_0_ = 1 is the refractive index of air, α is the divergence angle of the pump beam at the fiber output edge. For the Gaussian function in water, we have $G(x)=G_{0}\exp\left[-\frac{x^{2}}{2a_{w}^{2}}\right],$ where *a_w_* = *a* × (tanα_w_/tanα), α = arcsin(NA), α*_w_* = arcsin(NA/*m_w_*), *m_w_* = 1.33 is the refractive index of water. The estimates gave α*_w_* = 62 μm.

### 4.2. Profile of Nafion Distribution

To find the spatial distribution of *ρ*(*x*) during soaking of Nafion in water with various deuterium content, an inverse problem to the luminescence intensity distribution had to be solved. We used the following algorithm: we seek the function *ρ*(*x*) in the form
(9)$$\rho (x)=B\left[n^{\left(0\right)}(x-\xi )+\theta (x-\xi )\cdot b\exp\left(-q{\left(x-\xi \right)}^{2}\right)\right],$$
where, as earlier, *θ*(*x*) is Heaviside function. In this simple model, we represent the residual “solid” Nafion plate as a rectangle of fixed thickness *d* = 175 μm. The Nafion density inside this plate is taken to be proportional to $n^{\left(0\right)}(x)=\theta (x+d)-\theta (x)$. The fitting parameter *ξ* stands for the uncertainty in the shift of the Nafion–water boundary during swelling. This shift occurs on immersion of Nafion plate in water and does not exceed 30 μm. For all the liquids investigated, this uncertainty is close to the micrometric screw step, equal to 25 μm.

The search for fitting parameters *ξ*, *q*, *B* and *b* was performed through minimization of the discrepancy functional between the theoretical and experimental values of the function *P*(*x*). The greatest practical interest is the value of the parameter *q*. This parameter denotes the reciprocal of the variance of the Gaussian function that corresponds to the half-width of the Gaussian distribution $b\exp\left(-q{\left(x-\xi \right)}^{2}\right)$. After substituting Equation (9) into Equation (6) and integrating we obtain the distribution function *P*(*x*) in the form
(10)$$P\left(x\right)=A+G_{0}B\frac{a_{w}\sqrt{\pi }}{\sqrt{2}}\cdot \left(\begin{array}{l} \left[\mathrm{erf}\left(\frac{x+d-\xi }{\sqrt{2a_{w}^{2}}}\right)-\mathrm{erf}\left(\frac{x-\xi }{\sqrt{2a_{w}^{2}}}\right)\right]+\\ +\frac{b}{\sqrt{2a_{w}^{2}q+1}}\left[1-\mathrm{erf}\left(\frac{-x+\xi }{\sqrt{2a_{w}^{2}}\sqrt{2a_{w}^{2}q+1}}\right)\right]\exp\left(-\frac{q{\left(x-\xi \right)}^{2}}{2a_{w}^{2}q+1}\right) \end{array}\right).$$

## 5. Spatial Distribution of Nafion in the near Surface Region

In Figure 10 we present the experimental dependences of *P*(*x*) (black circles) and the corresponding theoretical *P*(*x*) curves (red color) for different deuterium content. The blue curves stand for the theoretical spatial distributions of Nafion density *ρ*(*x*), obtained in accordance with the parametric model described in Section 4. Since the total mass of Nafion is conserved when soaking, the dependencies of *ρ*(*x*) were normalized to the total area under this curve.

After soaking the height of the equivalent rectangle which stands for the effective density of the residual Nafion plate, naturally decreases. This is attributed to a reassembly of polymer to give conducting nanochannels filled with water inside the membrane. It is possible that the unwinding of Nafion fibers into the adjoining bulk liquid causes a reduction in the thickness *d* of the initial Nafion plate. To verify this, it would be necessary to measure the spatial distribution *ρ*(*x*) both to the right and to the left sides of the plate, and inside the plate. However, the design of the experimental setup does not allow us to do this. We expect that there should be a density gradient in the near-surface layer of the plate, i.e., the distribution of the Nafion density inside the plate is no longer uniform. However, a reliable determination of this non-uniformity is beyond our measuring capabilities. For this reason, and to facilitate analysis, the inhomogeneous density distribution inside the Nafion plate is replaced by a homogeneous distribution with an effective density that turns out to be lower than that of dry Nafion plate (Figure 10). This approach is justified by the fact that we are interested mainly in the formation of the EZ, which, according to Ref. [19], arises during the process of soaking and extends into the bulk liquid by hundreds of microns.

In Figure 11 we exhibit the dependence of the quantity *X*_0_ = (2*q*)^−1/2^; (recall that *q* is the value of inverse dispersion in the Gaussian function in Equation (10)). The abscissa value *X*_0_ is measured from the level of solid boundary *x* = 0, and the experimental points are connected by splines. As follows from this graph, the value of *X*_0_ depends non-monotonically vs. the deuterium content. Note that Equation (6) belongs to the class of Fredholm integral equations of the first kind [39]. It is known that these equations have a single solution, but this solution is unstable with respect to small deviations of the theoretical curve *P*(*x*) which approximates the experimental values. The solution to such an equation is technically an ill-posed problem. Thus, a question arises about the consistency between the distribution obtained for *ρ*(*x*) in the bulk liquid and the distribution, obtained from the left part of Equation (6); the latter can be measured in experiment.

It is important that the effect of EZ formation in DI water with deuterium content *C* = 157 ppm (Refs. [19,20,21]) implies that the characteristic size of the region, from which the colloidal micro-particles are pushed out, is approximately equal to 200–220 μm. It is straightforward to assume that this area is filled with polymeric fibers that unwound into the bulk water, but did not completely detach from the substrate. This model qualitatively represents the formation of the EZ. Our measurements have shown that the value of *X*_0_ does not change after a soaking time *t* > *t*′ = 30 min, i.e., the unwinding of polymer fibers into the bulk of liquid cannot be described by diffusion kinetics; however, the temporal dynamics of the unwinding process has not been studied.

These results are more than unexpected. They are extraordinary. The dependence of the EZ size on deuterium content is non-monotonic! The interpretation of these results awaits further exploration. We can assume that the relative hydration of sulfonic and fluoride groups that sets the intrinsic curvature (and self-assembly) of the polymer matrix must be determined by competitive adsorption between deuterium and water. Clearly deuterium “wins out” over water molecules. The polymer fibers are attached at one end to the membrane surface, i.e., no complete detachment and escape into the liquid bulk occurs. A quantitative modelling of this process can be found in Ref. [32].

It is very important to note that the particles of Nafion in bulk water do not tear off from the surface completely. There is a rearrangement that leads to a stationary near-surface profile of the polymer density in the liquid bulk. The corresponding distribution for the analogous glycocalyx–endothelial surface layer in physiology is a string of sparce polymer strands that stretch out normal to the GC surface about 1 μm. However, the scales are different here, with the Nafion EZ ~300 μm.

Topologically the arrangement may well be quite subtle. For example, it could involve entangled polymer particle clumps reminiscent of ribosomes for DNA winding, interspersed with linear stretches of polymer. When Nafion swells in DI water (the deuterium content is 157 ppm), polymer fibers unwind into the bulk liquid, and reassemble into an organization of strands different to that of the polymer bulk. This effect is absent in deuterium depleted water.

To throw light on this curious effect, an additional experiment was carried out. We measured the size distribution of Nafion nanometer-scaled particles in aqueous suspensions, prepared from both DI water and DDW, see Ref. [40]. To carry out this experiment, a Nafion plate (thickness 175 μm) was first cut into particles of about 1 mm in size, and then these particles were ground in an MSK-SFM-12M Milling Machine (MTI Corporation, Richmond, CA, USA) to form a polydisperse powder. The size distribution of Nafion particles in this powder in aqueous suspensions in DI water, and in DDW, were determined by dynamic light scattering (DLS) experiments with a Zetasizer Ultra/Pro setup (Malvern, UK). The dynamic light scattering technique was comprehensively described in monographs [41,42]. The DLS setup was equipped with a continuous wave He − Ne laser at a wavelength of λ = 633 nm (maximum intensity is 4 mW) and a temperature controller; the scattering angle was 173°.

Figure 12 shows the scattering intensity distributions from Nafion particles in the two suspensions: DDW (panel (a)), and DI water (panel (b)). Evidently, due to the milling, a polydisperse suspension is formed. It includes particles with hydrodynamic diameter in a range from hundreds of nanometers to microns. It is not possible to detect particles with diameters exceeding 6–7μm via dynamic light scatteringsee, e.g., [43]. So, these do not show up in the plots of Figure 12. The insets in the graphs show the effective hydrodynamic diameters of scattering particles. The ordinate axis shows the percentage of scatterers with an effective size measured. The fact that the Nafion particle sizes in the two experiments differ is due to their different polydispersity. Significantly, the size of particles in DI water (panel (a)) is several times larger than those in a suspension in DDW (panel (b)). At the same time, there is no reason to believe that the differences in the sizes obtained for suspensions of Nafion particles in DI water and in DDW are due to the difference in the sizes of the initial powder of dry Nafion particles that were added to the liquids under study. Contamination due to ions from the ball mill, we believe, are irrelevant. Therefore, we can claim that *the difference in the sizes of Nafion particles in the two aqueous suspensions can only be associated with the effect of unwinding of polymer fibers in DI water and the absence of this effect in DDW.*

We can hypothesize that deuterium competes with para- and ortho- forms of water around the SO_3_ groups. This changes the (probably chiral) nature of hydration along the polymer. This in turn affects the capacity of the polymer to wind up into a string of nanoparticles separated by linear segments. These hairy “balls of string”, like ribosomes on DNA, attract each other to form colloidal aggregates of different polydispersity. Since the only mechanism for increasing the particle size in DI water is associated with unwinding of the polymer fibers, it seems that we have confirmed in a direct experiment that the polymer fibers unwind in DI water and do not in DDW.

## 6. Nafion Swelling in a Finite Volume. Formation of Water-Free Cavity

In a further attempt to probe the nature of the EZ we posed this question: what happens if the Nafion plate is swollen inside the cell containing liquid, when the distance *L* between the windows of the cell is less than *X*_0_? Here, *X*_0_ is the maximum size of area occupied with unwound fibers upon swelling in an unbounded liquid, see Figure 11. More precisely, the size *X*_0_ should be compared with the length *l* = (*L* − *L*_0_)/2, where *L*_0_ is the thickness of the Nafion plate. If the Nafion plate swells in DDW, the unwinding of the polymer fibers does not occur, *X*_0_ ≈ 0 accurate to the experimental error, i.e., there should not be no features of interest. However, if *X*_0_ > *L*, then we should expect that the Nafion fibers growing out from the bulk Nafion surface will abut against the cell windows.

In this experiment the polymer membrane was soaked under constraints, different to those realized in a cell whose size exceeds *L*_0_ (e.g., a Petri dish). To see what happens, we explored the dynamics of unwinding of polymer fibers in a confined volume. A Fourier transform infrared (FTIR) spectroscopy technique was used. We worked with Nafion N117 plates (Sigma Aldrich, USA) with a thickness of *L*_0_ = 175 μm and an area of 1 cm^2^. The Nafion plates were soaked in Milli-Q water (DI water) with a resistivity of 18 MΩ⋅cm (measured immediately after the preparation) and deuterium content 157 ppm. We also studied swelling of Nafion in deuterium depleted water (DDW, Sigma Aldrich, USA), deuterium content 3 ppm. We found significant differences in the two cases, DI water and DDW, when the polymer swells in such a limited volume.

### 6.1. Protocol for FTIR Experiments: Instrumentation

The experimental protocol is described in detail in Refs. [44,45]. In these experiments we measured the transmittance spectrum *K= I/I*_0_ in the IR spectral range. Here *I*_0_ is the intensity of the radiation that falls on a cell containing the liquid sample and Nafion plate, and *I* is the intensity of transmitted radiation. In the FTIR experiments, we studied the spectral range of λ = 1.8–2.2 μm. Since radiation in the IR range is absorbed by water molecules, and the Nafion does not absorb in the spectral range 1.8–2.2 μm, the value of *K* should decrease upon soaking in water. This choice for that spectral range was made because the absorbance of water in this range is not so high (it is attributed to a combination of asymmetric valence and flexural vibrations of the H_2_O molecule, see [46]). This allowed us to register the transmitted signal behind the cell with water. The experiments were carried out on a FTIR spectrometer FSM 2201 (LLC Infraspek, St. Petersburg, Russia). This setup is illustrated in Figure 13.

The spectrometer had the following characteristics: the total spectral range is 370–7800 cm^−1^, spectral resolution is 1.0 cm^−1^, absolute error is ±0.05 cm^−1^. The moment of time, when the liquid was poured into the cell, set the reference time *t*. Each measurement included 15 consecutive records of the transmittance *K*. The results of measurements were averaged. The total time of measuring the transmittance *K* at a particular wavelength amounts to 40 s. taking into account the subtraction of the background absorption due to air humidity. The time interval between each measurements was 5 min. During these intervals, the cell was removed from the spectrometer and cooled down until reaching room temperature, i.e., all measurements were carried out at the same temperature.

The cell had windows made of CaF_2_. This material is transparent to IR radiation in the spectral range of interest. The roughness of windows was about 2.5–5 μm. Before each experiment, the windows were rinsed with Milli-Q water and then dried by a stream of chemically pure nitrogen. The distance *L* between the windows was varied from 180 to 220 μm with a step of 10 μm by using special spacers made of aluminum foil.

In Figure 14a–d we exhibit the cell with Nafion plate, filled with DI water/DDW. The distance between the windows of the cell *L* = 200 μm. Bearing in mind that the thickness of the Nafion plate is *L*_0_ = 175 μm, we find that in this case an area, having the size *l* = (*L* − *L*_0_)/2 = 12.5 μm, is occupied by unwound polymer fibers on each side of the Nafion plate. Panel (a) illustrates schematically the spontaneous formation of an empty cavity when the cell is filled with DI water, and the absence of the cavity when filled with DDW. Panel (b) shows a photograph of the cell immediately after filling with DI water; an empty cavity, centered behind the window, is clearly seen in this photo. Panel (c) shows a photograph of the cell immediately after filling with DDW; in this case, the cavity is absent. It was very important that the volume of liquid, poured into the cell, always exceeds the total volume of the cell. In fact, when we poured a liquid through the inlet, marked on panel (a) with arrow directed downwards, some amount of water always spilled out of the outlet, marked on panel (a) with arrow directed upwards. The inlet and outlet are marked in red on panels (b) and (c). With this filling protocol, gas bubbles do not form inside the liquid, and the entire volume of the cell (except for the cavity) is filled with water. After filling, the inlet and outlet holes were closed with Teflon inserts, but not tightly, i.e., there was a possibility for air to enter the cell.

We note that the cavity is absent when filling with DDW, see panel (c). The cavity absence in this case can be explained by the fact that for DDW there is no effect of the polymer fibers unwinding, see Figure 11 and Figure 12. So we can assert that the formation of the cavity must be a consequence of unwinding of the polymer strands. It seems that the cavity illustrated in Figure 14b is formed due to an instantaneous unwinding of polymer fibers into the bulk liquid. These fibers rest against the cell windows. This gives rise to local shear stresses, which cause a “squeezing” of water molecules from the areas between the fibers. Since the polymer fibers abut forcibly against the cell windows, the water molecules are pushed out very efficiently from the areas between the fibers due to local stresses as these fibers are initially hydrophobic. At the same time, the peripheral zone of the plate (see Figure 14a,b) is always in contact with water, and the polymer fibers in this peripheral zone keep swelling, which results in eventual collapse of the cavity. The rate of the collapse should depend on the area of Nafion plate covered with water. To deal with this, the Nafion plates were prepared in such a way that they had approximately the same size and shape, see Figure 14d.

Consider the bulk Nafion membrane. After exposure to water the membrane is highly conducting with respect to water molecules. Its nanostructure then can only be understood, if we consider it to be made of nanotubes of water surrounded by a helix with facing sulfonic moieties, and separated by a hydrophobic Teflon base of the polymer. On exposure to bulk water the helical tubes unwind in the opposite direction to form linear threads with sulfonic helices coating a hydrophobic interior. This situation is illustrated on a qualitative level in Figure 15, where the ideal (left-handed) double-twist configuration of neighboring molecules about a central chiral molecule is shown (reprinted from Figure 4.32, Ref. [47]). For simplicity, a square arrangement of molecules is assumed; in practice, the array is more likely to be hexagonal.

### 6.2. Transmission Spectrum in FTIR Experiments

According to the Lambert–Beer law (see, e.g., [48]), the intensity *I* of transmitted beam should obey the formula
(11)$$I=I_{0}\exp\left(-\kappa L\right),$$
where *κ* is the extinction coefficient, and *L* is the distance between the cell windows. In Figure 16 we show a typical example of a transmittance spectrum *K* for DI water, poured into a cell with the distance between the windows *L* = 180 μm. For the spectral IR range 1.8 < λ < 2.2 μm the minimum *K*_min_ is related to the wavelength λ = 1.93 μm. For short times of soaking *t* and small distances *L* we obtained for the transmittance *K* at the wavelength λ = 1.8 μm (transmittance coefficients *K* at the wavelengths λ = 1.8 μm and 2.2 μm will be hereafter denoted as *K*(λ = 1.8 μm) and *K*(λ = 2.2 μm)) that *K*(λ = 1.8 μm) ≈ 0.7. Upon increasing *t* and *L* the value of *K*(λ = 1.8 μm) decreases slightly, while *K*(λ = 2.2 μm) decreases more strongly. The decrease in the transmittance *K* at the wavelengths λ = 1.8 and 2.2 μm is basically due to the contribution from a more intense absorption band for water, centered at λ = 3 μm, see Ref. [46]. Since we are interested in the quantity |ln*K*_min_| at the wavelength λ = 1.93 μm, it makes sense to count down the value of *K*_min_ from the level which is the same to all spectrograms obtained. Indeed, since the dependence|ln*K*_min_| at *K*_min_< 1 is a very steep function, the inaccuracies in finding *K*_min_should result in large errors in finding|ln*K*_min_|. For this reason we put hereafter *K*(λ = 1.8 μm) = 0.7 for all spectra.

In Figure 17 we present the results of measuring |ln*K*_min_| for water vs. the distance *L* between the windows for particular values of *L* = 180, 190, 200, 210 and 220 μm. The results were averaged over five consecutive measurements. The minimum value of the distance *L* = 180 μm was chosen from the following considerations: this value is approximately equal to the thickness of the Nafion plate *L*_0_ = 175 μm. The choice of the maximum value *L* = 220 μm is due to the fact that in this case the absorbed energy of light is very high, and therefore the intensity *I* of the transmitted beam approaches zero, i.e., the results of measurements become meaningless. The dependence |ln*K*_min_| vs. *L* is well approximated by straight-line function |ln*K*_min_| = 0.027 + 0.019⋅*L*. Thus, from Equation (11) we obtain for the extinction coefficient *κ* ≈ 0.019 μm^−1^. The value of |ln*K*_min_| for dry Nafion was also measured. In this case, the absorption is due to water molecules encapsulated inside the nanometer-sized cavities in the bulk of polymer matrix (see Ref. [1]), i.e.,|ln*K*_min_| = *κ*⋅(*C_w_*)_0_*L*_0_, where (*C_w_*)_0_ is the concentration of water in dry Nafion; we obtain (*C_w_*)_0_ = 0.174.

We also measured the transmittance spectra *K* for Nafion swelling in water, with an interval of 5 min. In Figure 18 we present typical spectra, taken for 70 < *t* < 100 min for DI water with the distance *L* = 200 μm. It is seen that the value of *K* decreases smoothly upon soaking.

In Figure 19, we exhibit the transmittance spectra *K* for swelling the Nafion plate in DDW. In this case, the distance between the windows *L* = 200 μm, and the swelling times 0.5 < *t* < 25 min. Here, *t* = 0 is related to the moment of filling the cell with liquid and performing the very first measurement, which lasts about 15 s., i.e., the first graph corresponds to the time *t* ≈ 0.5 min after filling the cell. It is seen that the transmittance spectra *K* for swelling in DDW are almost identical for all times *t*. We also present here the spectrum of *K* for DI water, taken at *t* ≈ 0.5 min after filling the cell. This spectrum differs significantly from the spectra obtained for swelling in DDW. According to [33], the transmission spectra of water in the range 1.8 < λ < 2.2 μm are the same for the samples with the deuterium content 1–10^4^ ppm, i.e., the effect detected cannot be associated with various absorptivity of DI water and DDW in this spectral range. We associate the difference in the transmittance spectra for swelling Nafion in DI water, taken at *t* = 0.5 min after filling the cell, and the transmittance spectra for swelling Nafion in DDW, taken at 0.5 < *t* < 25 min, with the absence of the water-free cavity in the case of swelling Nafion in DDW.

Based on the transmittance spectra obtained, we can now find the dependence of water concentration *C_w_* in the cell, filled with DI water or DDW and containing the Nafion plate subject to soaking. The water concentration obviously depends on the swelling time *t* and the distance *L*. Hereafter we will study the concentration of water 〈*C_w_*(*t*)〉, averaged over the length *L*. Rewriting the Beer–Lambert law (Equation (11)) in the form:(12)$$I\left(t\right)=I_{0}\exp\left(-\kappa \int_{0}^{L}C_{w}\left(t,x\right)dx\right)\approx I_{0}\exp\left(-\kappa \langle C_{w}\left(t\right)\rangle L\right),$$

We obtain for water concentration in the spectral minimum of the transmittance *K*, taken at the wavelength λ = 1.93 μm, the formula $\langle C_{w}\left(t\right)\rangle =\frac{\left|\ln K_{\min}(t)\right|}{\kappa L}$. In Figure 20 we exhibit the dependences 〈*C_w_*(*t*)〉 for the cell with the distance between the windows *L* = 200 μm. The dashed line is related to the concentration of water in dry Nafion; in this case (*C_w_*)_0_ = 0.174 (baseline).

In Figure 21 we show the dependence of 〈*C_w_*(*t*)〉 for swelling of a Nafion plate in DI water. The distance *L* between the windows of the cell was equal to 200 μm. In this case the empty (water-free) cavity arises, but does not completely disappear after ~200 min of swelling. In this Figure, we exhibit the graphs 〈*C_w_*(*t*)〉 for water samples, obtained from different Milli-Q devices, and recorded on different days. As seen in this figure, it makes no sense to calculate experimental errors within the first 100 min of swelling: the points on the graphs actually coincide, and only at the times *t* > 100 min does the difference between the extreme points along the ordinate axis (the scatter in the dependence 〈*C_w_*(*t*)〉) exceeds 1% of the values 〈*C_w_*(*t*)〉 at these times. This is why in Figure 21 we present the experimental errors only for the times of swelling *t*> 100 min. The values of these errors were calculated based on the results of five consecutive measurements. We can say that for DI water within the first 100 min of soaking, the 〈*C_w_*(*t*)〉 dependence remains rectilinear.

We also have studied cells with a distance *L* between windows in the range of 180–220 μm. In addition, we studied cells whose windows had different degrees of roughness. For rougher cell windows, the time of the cavity collapse was significantly higher than for windows with high quality of polishing. These results are described in our recent papers [44,45]; we did not include these results so as not to overload the text.

Summarizing this subsection, we can assert that when the Nafion membrane swells in DI water or DDW inside a cell, the volume of which is comparable to the volume of the membrane itself, the regimes of swelling in these liquids are quite different. The peculiarities of swelling in DI water are revealed by the constraints imposed by the cell windows. If the thickness of the cell (the distance *L* between the windows) is much less than the size of the area of “unwinding” of polymer fibers, that shows up in the experiments with photoluminescence spectroscopy (previous section), then the initially hydrophobic polymer fibers inevitably abut against the windows of the cell. This results in the appearance of a field of mechanical stresses and deformations between the cell windows and the unwound fibers. It is important that these stresses arise in the system of “twisted” polymer fibers, which initially have hydrophobic properties, i.e., the location of the water molecules, caught in the gaps between the unwound fibers, appears to be unstable. Due to the enhancement of the hydrophobic effect caused by local stresses, the water molecules located between the polymer fibers are pushed out, and this leads to the formation of an empty cavity between the membrane surface and the cell windows.

## 7. Specific Electrolyte (Hofmeister) Effects

### 7.1. Preliminaries

So far, we have considered samples of pure water which differ only in deuterium content: deuterium depleted water (DDW) 3 ppm and 157 ppm (DI water). From Figure 21, we can see that when the Nafion membrane is soaked in DDW, a cavity, which is free from water, is not formed due to the absence of polymer fiber unraveling. This is a convincing hint that deuterium affects chirality as discussed above. We also have studied LiCl, NaCl, KCl and CsCl solutions in DI water at concentrations of 10^−3^–1 M. We have not performed FTIR spectroscopy experiments to study salts solutions based on DDW. Indeed, in the experiments with FTIR spectroscopy we study the kinetics of the cavity collapse that occurs due to the unwinding of polymer fibers. Since this cavity does not occur in the case where Nafion is soaked in DDW, it makes no sense to study the dynamics of cavity collapse in DDW-based salt solutions. However, in the case of DI water, we can expect that ionic specificity will show up during the cavity formation. In Figure 22 we shows the results of measurements of the 〈*C_w_*(*t*)〉 value for aqueous solutions of LiCl, NaCl, KCl and CsCl for a concentration of 0.1 M and for soaking times 0 < *t* < 100 min. Experimental points are averaged over sixmeasurements; confidence intervals are indicated in this figure. In addition, reference data for DI water arealso given; in this case, confidence intervals are not presented, since for DI water the experimental points with high accuracy lie on a straight line, see Figure 21.

We see that for all the salts studied, the run of the experimental dependences is not rectilinear, as in the case of DI (reference) water. Thus, deviations from the straight-line dependence can be due to the addition of foreign ionic impurities. Apparently, in experiments with FTIR spectrometry it is impossible to obtain more detailed information about the role of ionic additives, and especially about the manifestation of specific ion effects during the formation of a water-free cavity.

### 7.2. Salt Effects via the Second Protocol

The effects of electrolytes on the EZ due to unwinding of polymer fibers were therefore studied via a different technique. In Figure 23 we exhibit a schematic of the experimental setup for photoluminescent spectroscopy. This is the second protocol for our experiments.

In this second experimental protocol, contrary to the first one, the Nafion plate was not shifted with respect to the optical axis: its position remained fixed during the entire experiment. As in the first protocol, the plate was fixed in parallel to the optical axis, i.e., the geometry of grazing incidence of the pump wave was kept. The description of the second experimental protocol is similar to that of the first protocol, see the comments to Figure 8. Note that the experiments performed with this protocol allowed us to study the dynamics of the luminescence intensity as a function of the time *t* of soaking the Nafion plate in the test liquid. 

The start for soaking time (*t* = 0) corresponds to the moment of pouring the liquid sample into the cell. The only difference from the experiments performed according to the first protocol, described above, is that the Nafion plate was fixed along the optical axis during the entire experiment and was not shifted; in this case, the stepper motor was used only for fine adjustment of the plate position relative to the optical axis.

Assuming that the luminescence centers are terminal sulfonic groups, whose volume number density *ρ*(*t*) in the boundary layer decreases exponentially (see Equation (5)), and bearing in mind that the luminescence intensity at the spectral maximum (λ = 508 nm, see Figure 4) *I*(*t*) ∝*ρ*(*t*)⋅*σ_lum_*, the value of *I*(*t*) in the case of *σ_lum_ = const* also obeys an exponential decay law. In Figure 24 we exhibit the luminescence intensity *I*(*t*) vs. the time *t* of soaking in DI water and in DDW.

It is seen that the dependences do follow an exponential decay. The pre-exponential factors, and relaxation times are similar for both liquids. So, we can deduce that specific ion effects arising from the Nafion surface will not show up in our photoluminescence experiments when the polymer is soaked in water with different electrolytes.

In what follows, we show the results of measurements of luminescence intensity *I*(*t*) vs. the soaking time. The measurements were made for LiCl (Figure 25), NaCl (Figure 26), KCl (Figure 27) and CsCl (Figure 28) solutions with salt concentrations of 0.1, 0.17 and 1 M, in DI water and DDW.

The choice of concentrations is because of the very different behavior of electrolytes above and below the critical physiological concentration of 0.17 M. If in the concentration is below 0.17 M, gas bubbles fuse, see Refs. [49,50,51,52]. Above the critical 0.17 M the colloidal suspension of gas bubbles is stable, and the bubbles do not fuse. This remarkable phenomenon is not explained by current theories of physical chemistry. It is which is crucially in biology for oxygen and nitrogen uptake, and CO_2_ expulsion. see Refs. [34,35] and [53,54,55,56]. The results, reported in Figure 25, Figure 26, Figure 27 and Figure 28, are extraordinarily complicated.

For CsCl solutions based on DI water and DDW at concentration of 0.17 M the *I*(*t*) dependence is not exponential, but exhibits an oscillation the first 25 min of swelling.

The luminescence intensity vs. swelling time *I*(*t*) decays exponentially, see the insets in the figures. The luminescence intensities reflect the depletion of Nafion at the surface as a function of time. This is a measure of the growth of the EZ, i.e., the unraveling of the polymer as it expands into bulk electrolyte–deuterium–water solution. The asymptotes of the curves (the free constants Y_0_, i.e., values of exponential functions at *t* → ∞, given in the insets) are an indication of how much the Nafion surface has been depleted and forms the EZ. The values of (Y_0_)_DI_water_, (Y_0_)_DDW_ and their ratios for each particular concentration *C* are listed in Table 1, Table 2 and Table 3.

The results are extraordinary. They must reflect several competing specific factors that are not easy to make sense of. There is no identifiable change that would reflect the bubble–bubble fusion inhibition phenomenon, see Refs. [49,50,51,52], or the corresponding critical nanobubble formation that occurs above 0.17 molar for 1-1 electrolytes, see [53]. There are remarkable differences in swelling between Li and Na on the one hand, and K and Cs on the other. These Hofmeister effects probably reflect the qualitative differences in binding of the cations to the sulfonic groups. Tight binding alone could and may produce a Manning polyelectrolyte condensation into ribosome-like entities along the strands, see Ref. [57].

What is even more extraordinary are the differences between DI water and deuterium depleted water. The only way we can make sense of that is in terms of the discussion on Section 1.1 and effects of deuterium variation in Section 2.

Deuterium adsorbs at sulfonic groups and displaces the competing para and ortho forms of water for its hydration. That allows the unravelling Nafion fibers to assemble into helices with a hydrophobic interior and hydrophilic sulfonate ions covering unraveled polymer strands in the salt solution. These nanocylinders are the reverse of those that form inside the bulk Nafion. The interior conducting nanohelices have hydrated cores surrounded by the fluorinated bulk polymer.

### 7.3. Parallels between the EZ and the Glycocalyx in Physiology

We summarize: On soaking, the Nafion appears to assemble into conducting nanotubes, helices with sulfonic groups of the polymer surrounding water, and a hydrophobic teflon-like exterior. It would be impossible for these hydrophobic nano-cylinders to unravel and penetrate the bulk water. More probably the cavity formation is due to an unravelling via a phase change. This can happen by an inversion of the hydrophobic polymer cylinders into charged hydrophilic helices that surround a hydrophobic core. This can then expand into the bulk liquid.

A first question that occurs is this: how it is possible for a strand of Nafion polymer to unwind up to the size of the EZ, which amounts to several hundreds of microns, (see Ref. [19]). An analogy from biology suggests it really is highly possible. The DNA molecule unwinds and rewinds to pack inside the cell nucleus with extravagant ease and reproducibility, while the length of a single macromolecule can be a meter or so, see, e.g., Ref. [58]. This suggests it is possible for polymeric strands at the Nafion surface to unwind, reorganize and to span the large width of the EZ.

A second question is how is it also possible for such a stretched-out nanostructure to form a rigid parallel palisade of wire-like threads, which can repel colloidal particles?

Again, there is a similar phenomenon in the glycocalyx and its own EZ, the endothelial surface layer. As was shown in a recent paper [59], several bacterial strains, including pathogens associated with hospital infections and/or foodborne outbreaks, are efficiently ejected from the EZ near the surface of Nafion. At the same time, as shown in Refs. [34,35], the ESL is an analogue of the EZ of Nafion in the sense that ESL repels colloidal particles over micron distances, similar to red blood cells and bacteria.

The glycocalyx (extra-cellular matrix, see Refs. [34,35,60] for more detail) can be considered as the analogue of unwound Nafion fibers, since the glycocalyx is built on top of the cell lipid membrane and as much a 50 nm thick. This is much larger than the thickness of the cell phospholipid lipid membrane, ~3 nm. The glycocalyx consists predominately of 50–90% heparin sulfate, and the remaining 10–50% is dermatan sulfate, chondroitin sulfate, keratin sulfate and hyaluronan, see, e.g., [61]. The differences from Nafion are that the glycocalyx is made of sugar-like (sialic acid) polymers plus sulfate. Note that the fluorocarbon group of Nafion is essentially more hydrophobic than the hydrocarbon-oxygen groups of the glycocalyx. So, while this is a big difference, the phenomena the Nafion-EZ and GC-ESL exhibit are so similar that they warrant comparison.

Extending out from the glycocalyx is the endothelial surface layer.The width of the endothelial surface layer (the “biological EZ”) is about a micron, an enormous distance compared with the glycocalyx itself) ~50 nm and the cell membrane ~3 nm.Its nature has only been revealed recently. It comprises very dilute hydrated polymer strands teased out from the glycocalyx that are separated by a foam of carbon dioxide nanobubbles. These nanobubbles are lethal to pathogens, see Refs. [34,35].

### 7.4. Very Long Range Fluctuation Forces (Technical)

There are differences between the ESL and EZ of Nafion (EZ, see Ref. [19]): the ESL exists and is bounded by the bloodstream, which is the aqueous salt solution at physiological concentration, equivalent to 0.17 M for NaCl solution. It is fed and replenished by carbon dioxide nanobubbles from metabolism emerging from within the tissue, see [34,35] for more detail. The result is a dynamic foam bounded by dilute polymeric strings perpendicular to the surface, see Figure 29). We emphasize once again that both the ESL and EZ repel colloidal particles, bacteria, T-cells, red blood cells and so on, and the ESL protects the underlying tissue. The differences in scale are big. About 1 μm for the ESL and around 300 μm for the EZ. The ESL formation is essentially the same phenomenon as for Nafion, but in lower key. The ESL is probably a helical assembly formed of two or more of its constituent polymers. For both EZ and ESL the parallel array of the linear nanostructured polymers is difficult to understand. It might mean that the EZ also contains nanobubbles of dissolved air.

However, there is another phenomenon that is operating. There are very strong long-range non additive molecular forces, which act between oriented in parallel to one another and conducting thin cylinders. Conduction effects give rise to a much longer-range attractive force than the usual *R*^−4^ van der Waals potential result for non-conducting rods. With polyelectrolytes and conducting cylinders long range forces arise from correlation of current fluctuations within or along the cylinders. The force between the cylinders is attractive and proportional to $R^{-3}{\left[\ln\left(R/a\right)\right]}^{-3/2}$ per unit length, see Refs. [63,64,65,66]. Here *R* is the distance between rods and their radii. If they interact via a screened Coulomb potential, this force is proportional to *R*^−3^. The possibility of such long-range forces was first mooted by London and has been quantified 50 years ago. As well as currents due to electronic fluctuations it is possible in biological systems to encounter currents due to proton or other ionic fluctuations. For example, in our cases the polymers have ionizable groups on the surface. Fluctuations in the counterions (the electric double layer) of dissociated sulfonic groups will give rise to currents along the surface. For a cylindrical molecule such as DNA or a virus this region may take the form of a sheath with effective thickness twice the Debye screening length. Two-cylinder interactions are attractive, three-cylinder forces are repulsive and may be of considerable magnitude. This peculiar fact explains why DNA is two stranded. With an array these strong many body forces provide the required rigidity to compression.

## 8. Summing Up

The extensive near surface region of the water–Nafion boundary stretches for up to 300 microns. The nature of this mysterious EZ has presented a challenge for several decades. It repels colloidal particles and biological cells and even proteins. It is so mysterious and unlike anything else seen in polymer science that it invited the controversial, but apparently reasonable appellation “4th phase of water”. At the same time, the possibility that the nanostructure of the conducting bulk Nafion might be other than an amorphous tangled polymer seems not to have been considered. That is an important pragmatic question in view of the use of Nafion in a fuel cell.

We have reviewed a range of previous experiments and reported a range new of experiments designed to probe the EZ via novel laser photoluminescence spectroscopy.

The results are astonishing. DI water contains 157 ppm deuterium. It exhibits the familiar EZ. However, there is no EZ in deuterium depleted water! The nature of the EZ and its extension varies considerably with the amount of deuterium added.

We have studied the EZ and its evolution as a function of time for both deuterium depleted water and DI water for alkali halide chloride salts Li^+^, Na^+^, K^+^ and Cs^+^. The studies embraced a range of concentrations above and below the critical concentration for nanobubble stability.

The differences between Li^+^ and Na^+^, and especially between Na^+^ and K^+^ are large and absolutely unexpected.

In a subsequent publication we will report an extensive set of experiments on amino acid interactions with the Nafion membranes. These provide clues to how signaling proteins between cells and bacteria via specific proteins work.

## Figures and Tables

**Figure 1 polymers-15-02214-f001:**
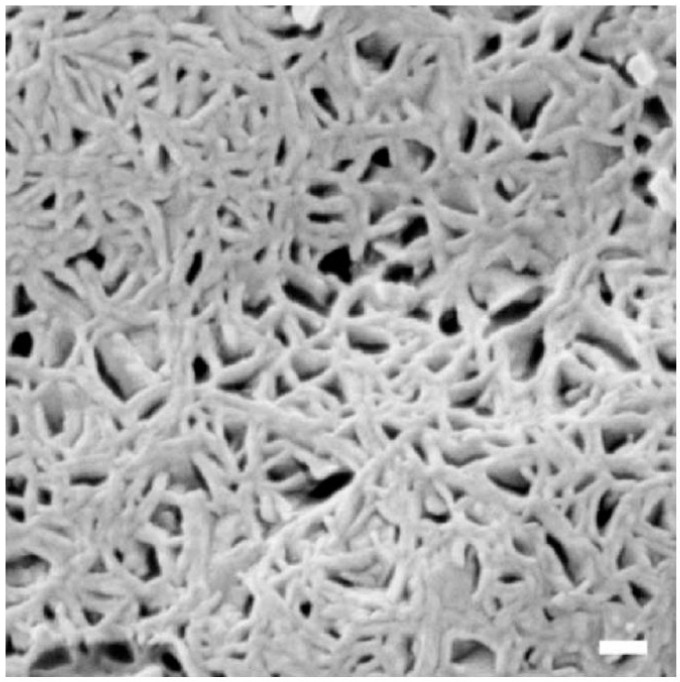
Cryo-SEM micrograph of an iso-octane/DDAB surfactant/water microemulsion. Component ratios by weight are (48:32:20) microemulsion. The bar corresponds to 100 nm. Reprinted from [16] with permission.

**Figure 2 polymers-15-02214-f002:**
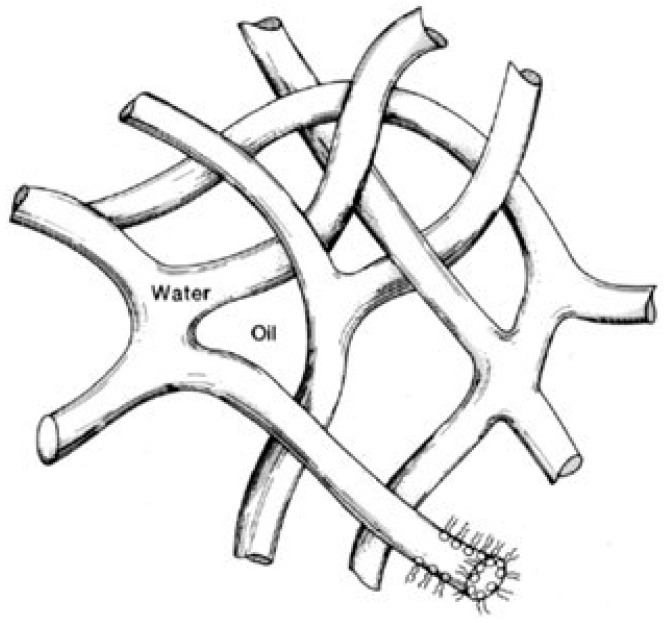
Details of the microstructure of micro emulsion formed from double chained cationic surfactant didodecyl dimethyl ammonium bromide (DDAB) water and alkane. Multiply connected nanotubes of water are coated with surfactant. These structures mimic the helical water tubes that form in bulk swollen Nafion. Adapted from [17,18] with permission.

**Figure 3 polymers-15-02214-f003:**
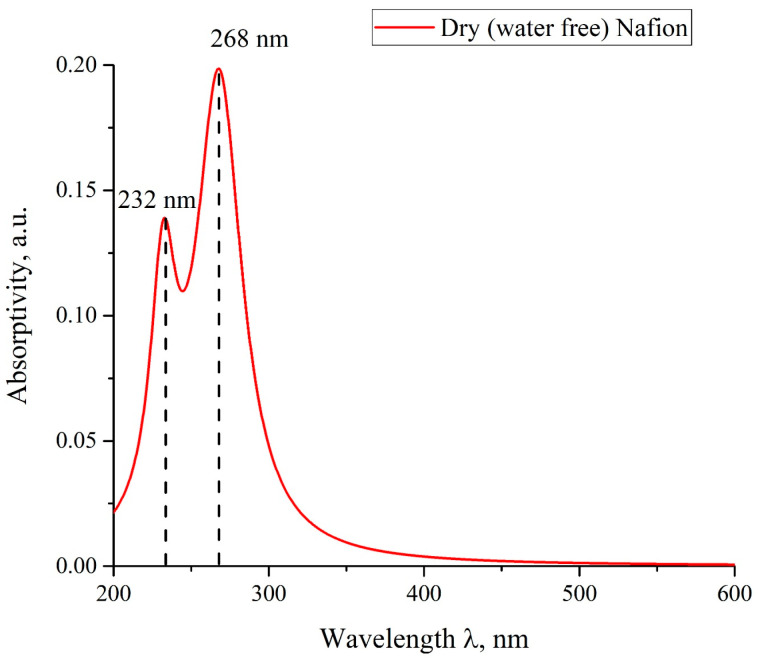
The absorptivity spectrum for dry Nafion.

**Figure 4 polymers-15-02214-f004:**
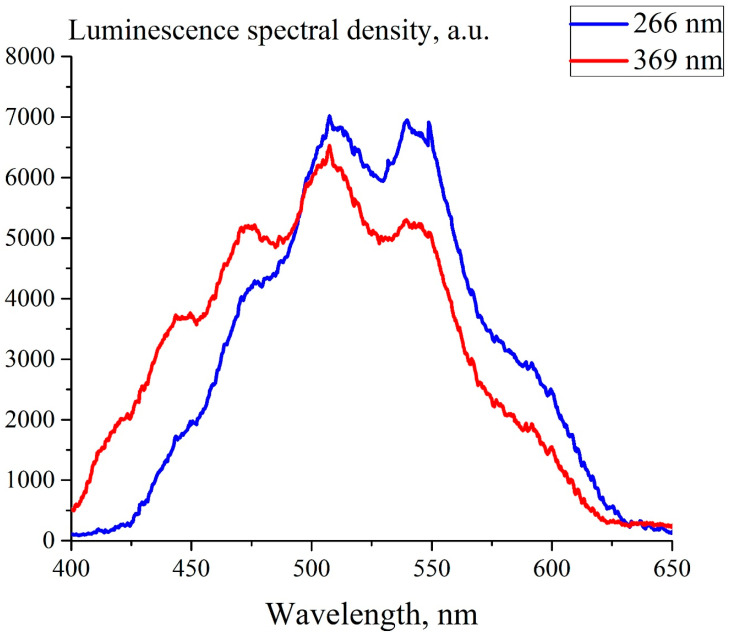
Luminescence spectra of dry Nafion irradiated at the wavelengths λ = 266 nm (blue curve) and 369 nm (red curve).

**Figure 5 polymers-15-02214-f005:**
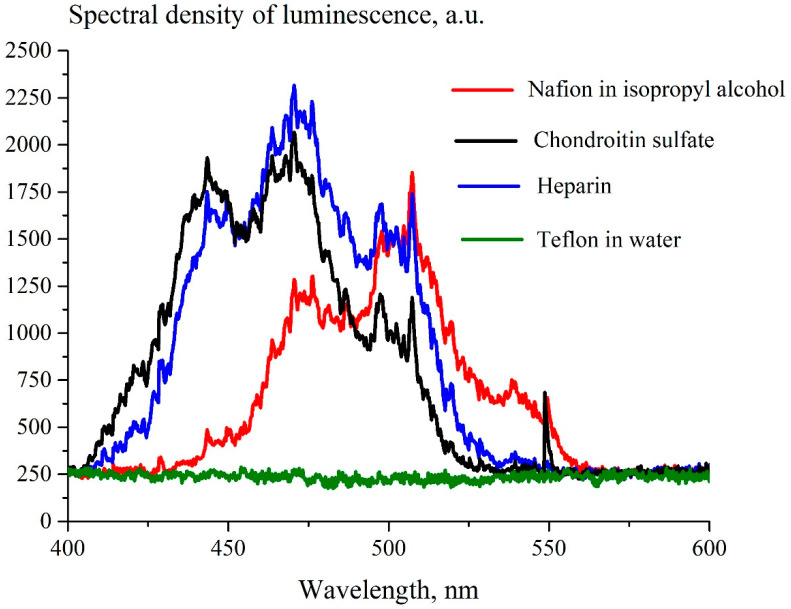
Luminescence spectra of Nafion, dissolved in isopropanol (red curve), aqueous solution of heparin sulfate (blue curve), aqueous solution of chondroitin sulfate (black curve) and Teflon in water (green straight line).The wavelength of the optical pump λ = 369 nm.

**Figure 6 polymers-15-02214-f006:**
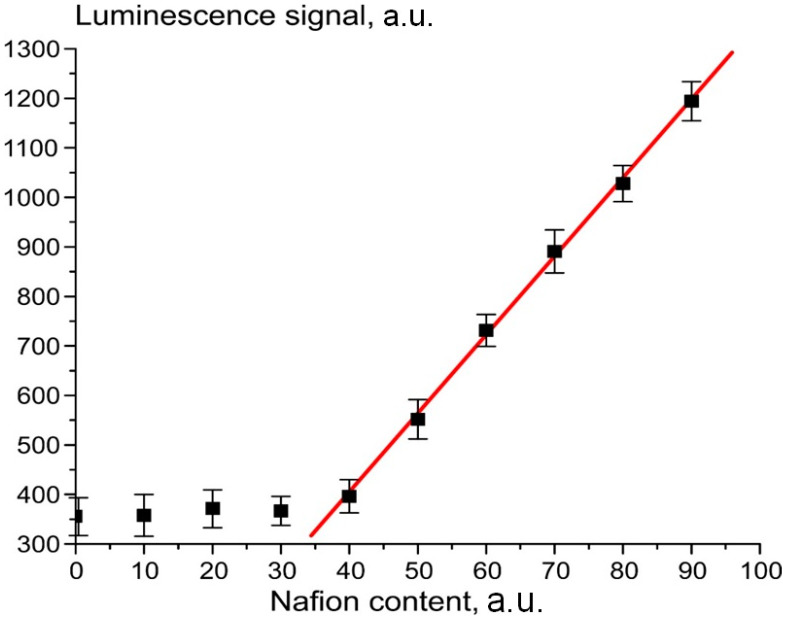
Dependence of the luminescence intensity in its spectral maximum vs. the content of Nafion, dissolved in isopropanol.

**Figure 7 polymers-15-02214-f007:**
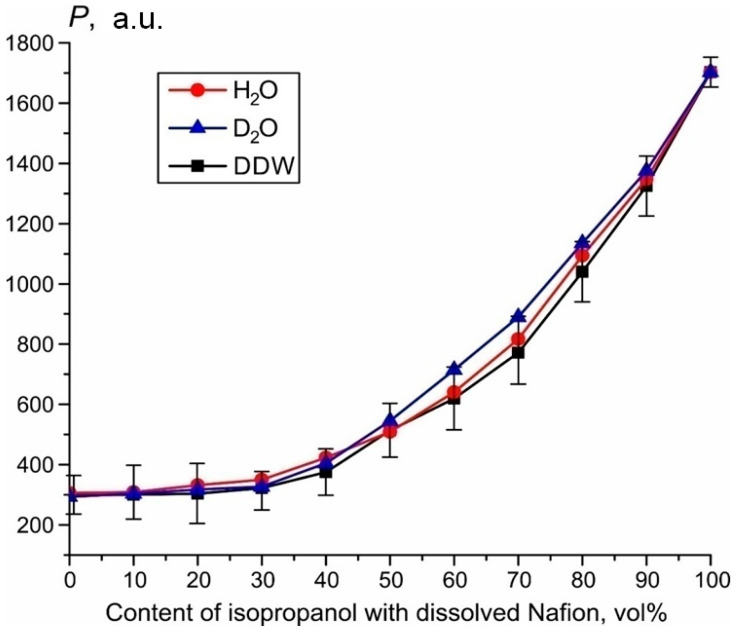
Luminescence signal P when Nafion solution in isopropanol is diluted with DDW (black curve), DI water (red curve) and D_2_O (blue curve).

**Figure 8 polymers-15-02214-f008:**
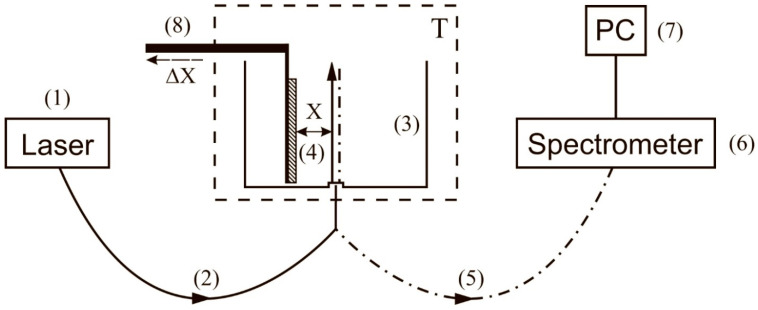
Schematic of the experimental setup for laser photoluminescence spectroscopy; first protocol. (1)—Laser diode (optical pumping); (2)—multimode optical fiber for transferring the pump radiation; (3)—cell for liquid samples; (4)—Nafion plate; (5)—multimode optical fiber for transferring the luminescence radiation; (6)—minispectrometer; (7)—computer; (8)—stepper motor for fine adjusting the position of the Nafion plate with respect to the optical axis.

**Figure 9 polymers-15-02214-f009:**
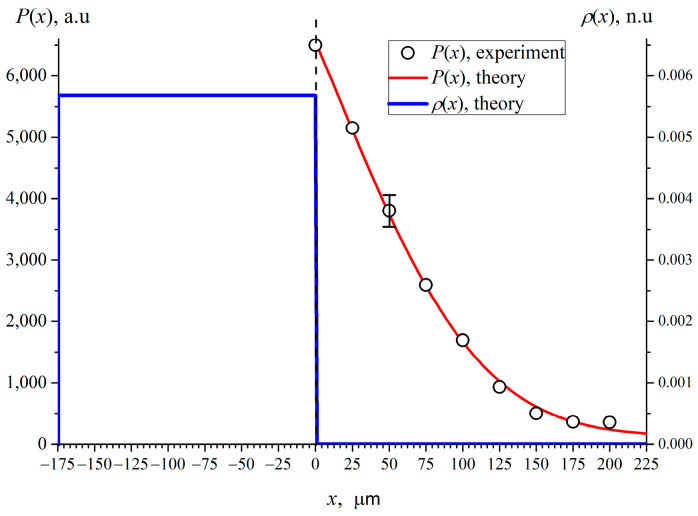
Dependence of the luminescence intensity *P*(*x*) for dry Nafion. Circles are the experimental points, the red curve is the theoretical approximation to the experimental dependence, the blue line is the normalized distribution of *ρ*(*x*).

**Figure 10 polymers-15-02214-f010:**
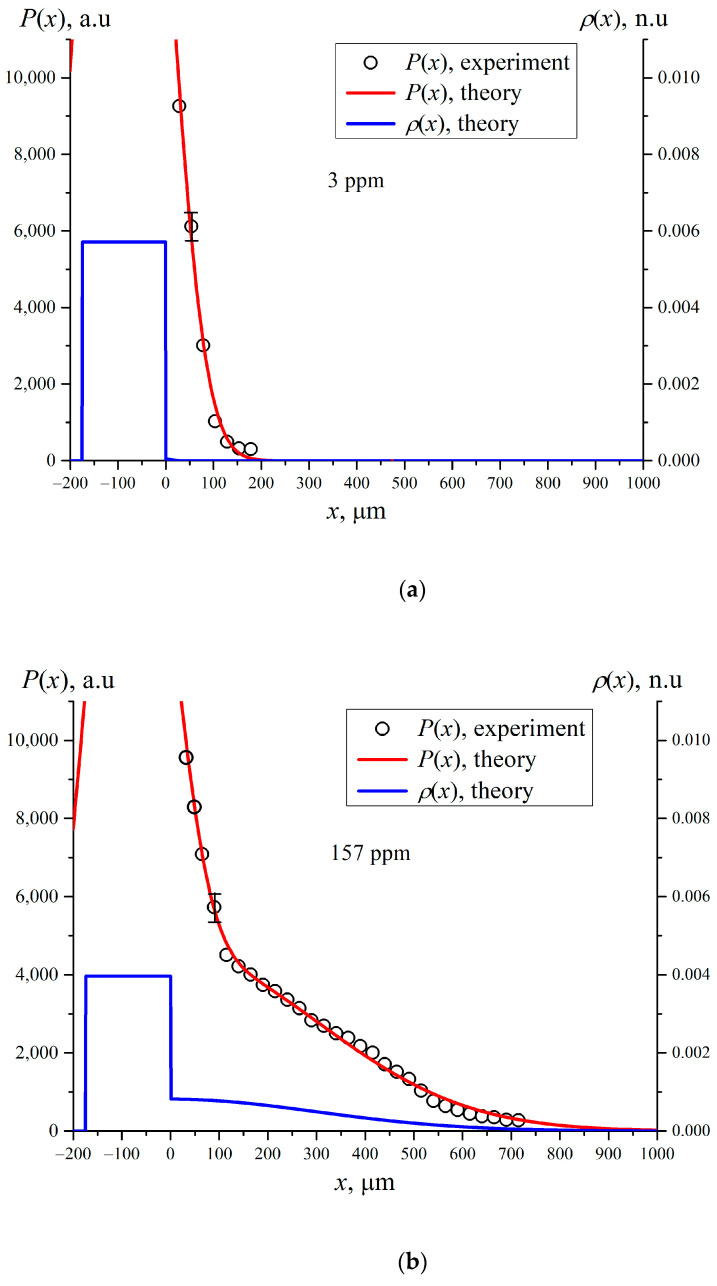
Dependencies of *P*(*x*) and *ρ*(*x*) for Nafion swollen in water with different deuterium content. The black circles are related to the experimental points *P*(*x*), the red curves represent the theoretical approximation, the blue curves are the theoretical density distribution *ρ*(*x*) of Nafion in the liquid bulk. Panel (**a**) is related to deuterium content 3 ppm (DDW); Panel (**b**) is related to deuterium content 157 ppm (DI water); Panel (**c**) is related to deuterium content 10^3^ ppm and panel (**d**) is related to deuterium content 10^6^ ppm (D_2_O).

**Figure 11 polymers-15-02214-f011:**
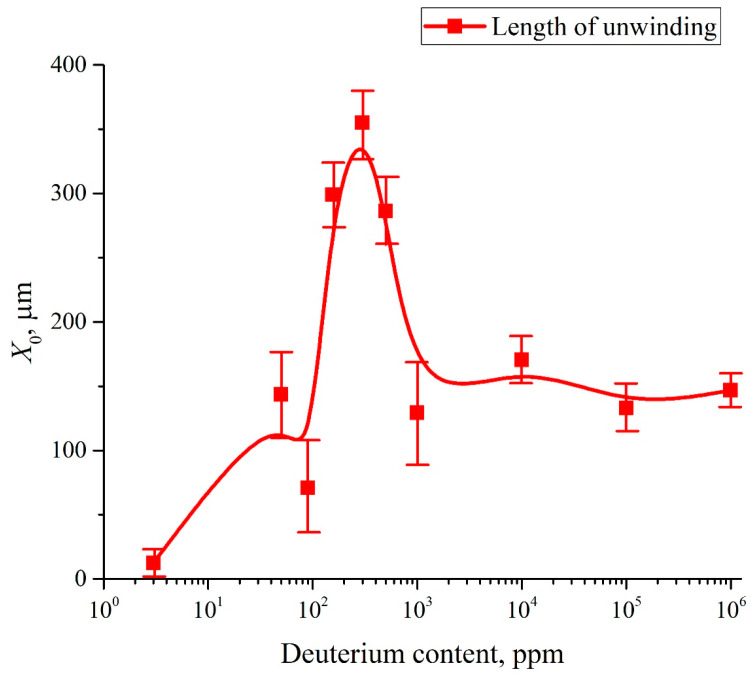
Dependence of the size of the unwound area in the bulk liquid vs. deuterium content.

**Figure 12 polymers-15-02214-f012:**
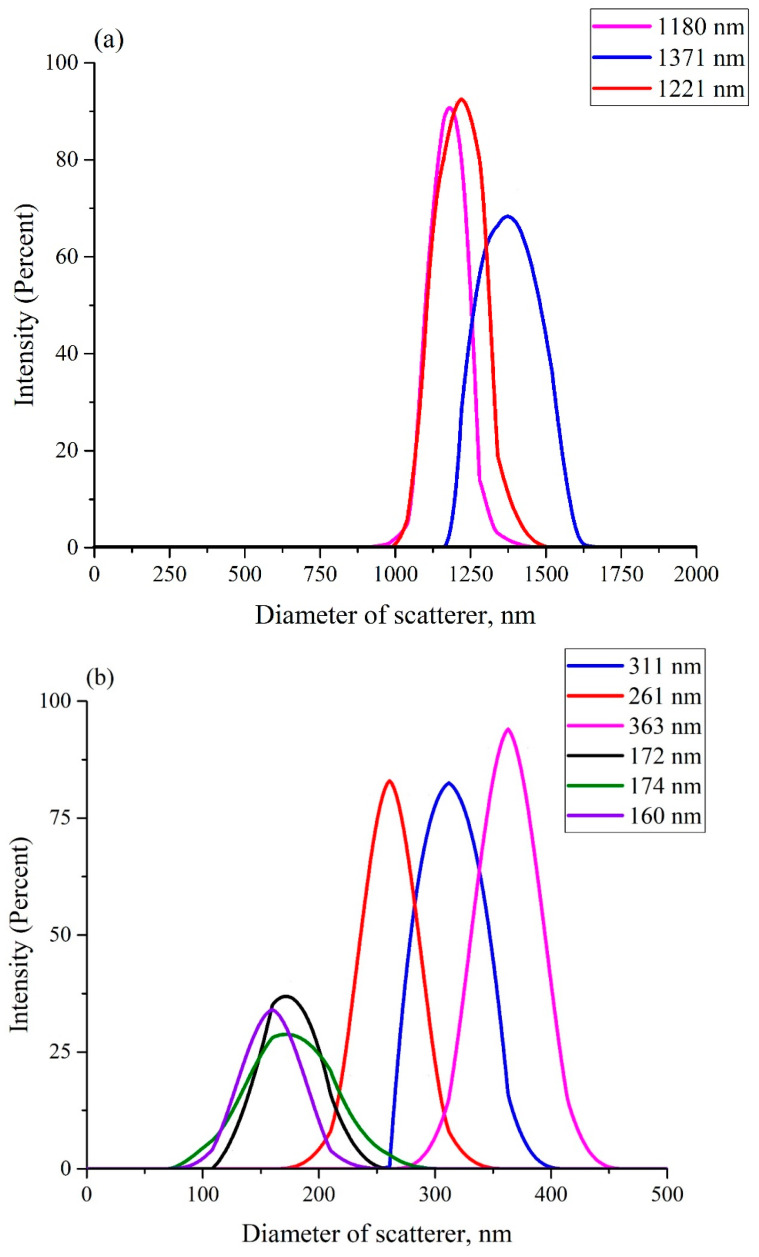
Histograms of hydrodynamic diameters of scatterers in DLS experiment. Panel (**a**)—DI water. Panel (**b**)—deuterium-depleted water.

**Figure 13 polymers-15-02214-f013:**
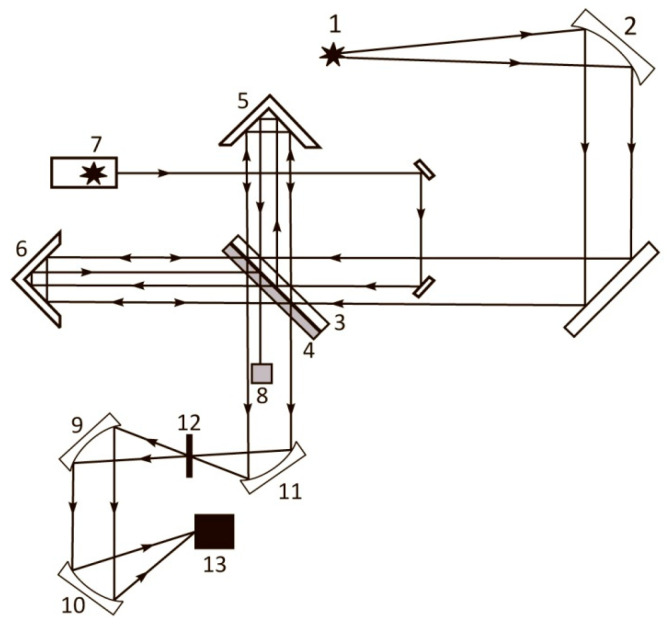
Schematic of Fourier spectrometer FSM 2201. 1—Source of IR radiation; 2, 9, 10, 11—Off-axis parabolic mirrors; 3, 4—Beam splitter and compensator (transparent in the IR range); 5—Fixed (stationary) reflector; 6—Movable reflector; 7—He-Ne laser; 8—Receiver of laser radiation; 12—Cell with liquid sample; 13—IR receiver.

**Figure 14 polymers-15-02214-f014:**
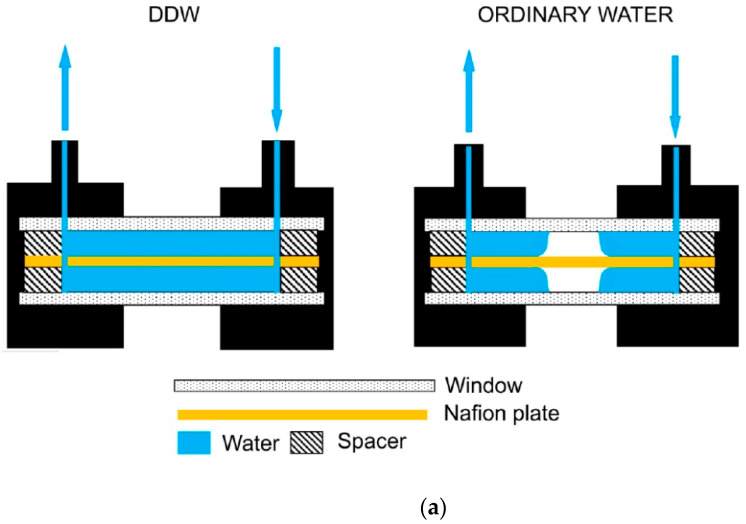
The cell used in our FTIR spectrometry experiments. Panel (**a**)—Schematic drawing of the cell design for filling with DDW and DI water. Panel (**b**)—Photo of the cell immediately after filling with DI water. The inlet/outlet holes are highlighted in red. The empty cavity is clearly seen. Panel (**c**)—Photo of the cell immediately after filling with DDW. The inlet/outlet holes are highlighted in red. The empty cavity is absent. Panel (**d**)—Photo of the Nafion plates used in FTIR experiments. The Nafion plates, used in our experiment, are approximately the same.

**Figure 15 polymers-15-02214-f015:**
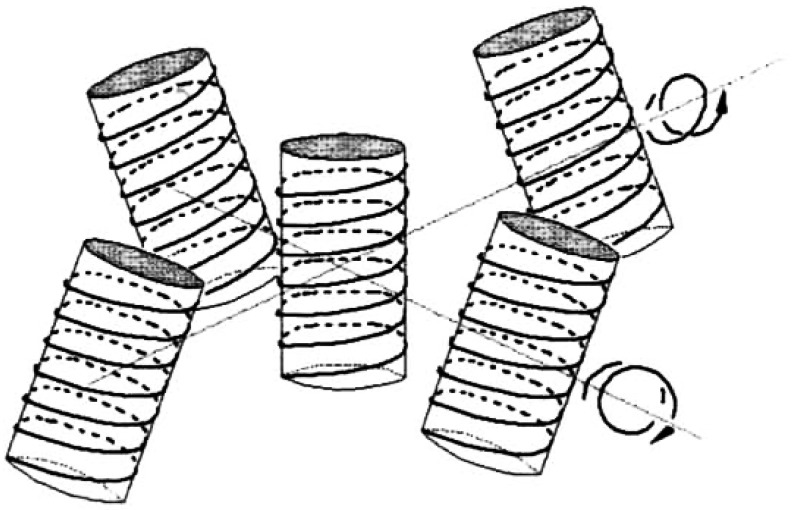
The ideal (left-handed) double-twist configuration of neighboring molecules about a central chiral molecule. For simplicity, a square arrangement of molecules is assumed; in practice, the array is more likely to be hexagonal (adapted from Figure 4.32, Ref. [47] with permission).

**Figure 16 polymers-15-02214-f016:**
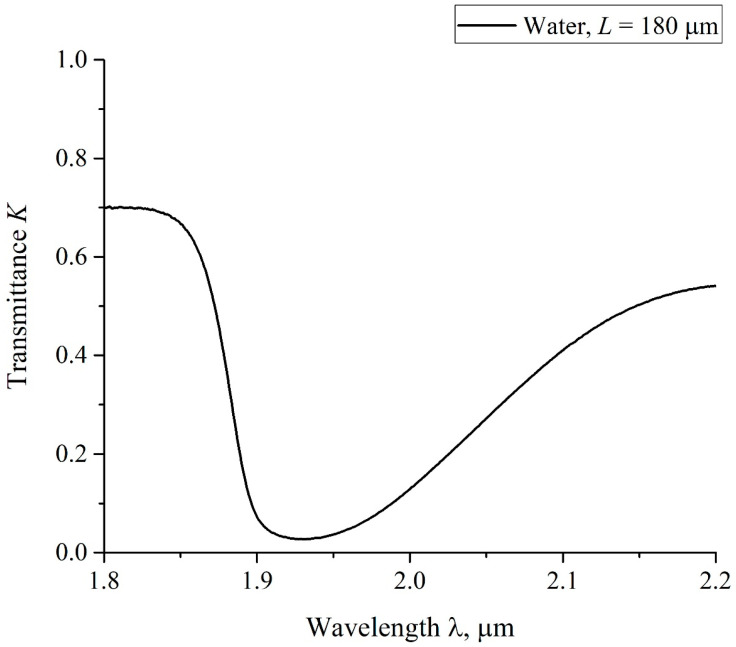
The transmittance *K* for water in the spectral range 1.8 < λ < 2.2 μm; the distance between the cell windows *L* = 180 μm.

**Figure 17 polymers-15-02214-f017:**
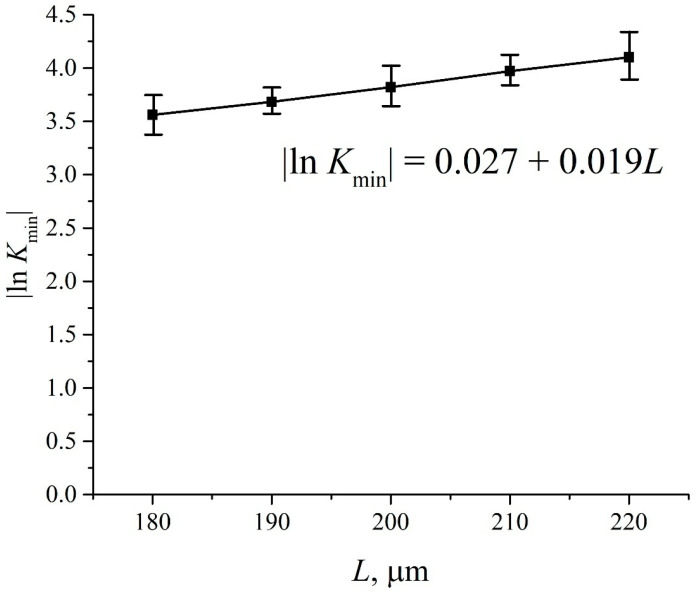
Dependence |ln*K*_min_|, taken at the wavelength λ = 1.93 μm, vs. the distance *L* between the cell windows for DI water. The value of *K*_min_ is related to the spectral minimum of the graph in Figure 16. The dependence |ln*K*_min_| vs. *L* is well approximated by function |ln*K*_min_| = 0.027 +0.019⋅*L*.

**Figure 18 polymers-15-02214-f018:**
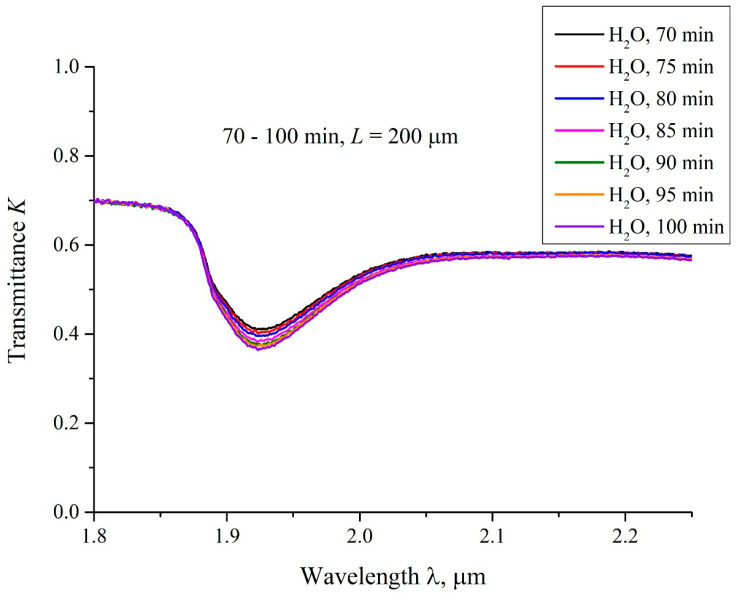
The transmittance spectra in the range 1.8 < λ < 2.2 μm for the case of Nafion swellingin DI water; the distance between the windows *L* = 200 μm, the curves are related to the swelling times *t* = 70, 75, 80, 85, 90, 95 and 100 min.

**Figure 19 polymers-15-02214-f019:**
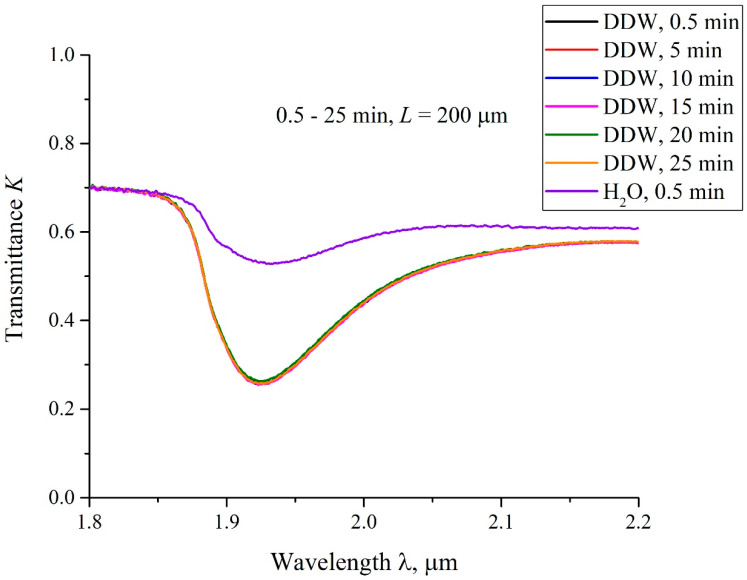
The transmittance spectra *K* in the IR range 1.8 < λ < 2.2 μm for the Nafion plate swelling in DDW in the cell, having the distance between the windows *L* = 200 μm. The graphs are related to the times of swelling *t* = 0.5, 5, 10, 15, 20 and 25 min. The upper curve is related to the transmittance spectrum *K* for DI water, the time of swelling is equal to 0.5 min.

**Figure 20 polymers-15-02214-f020:**
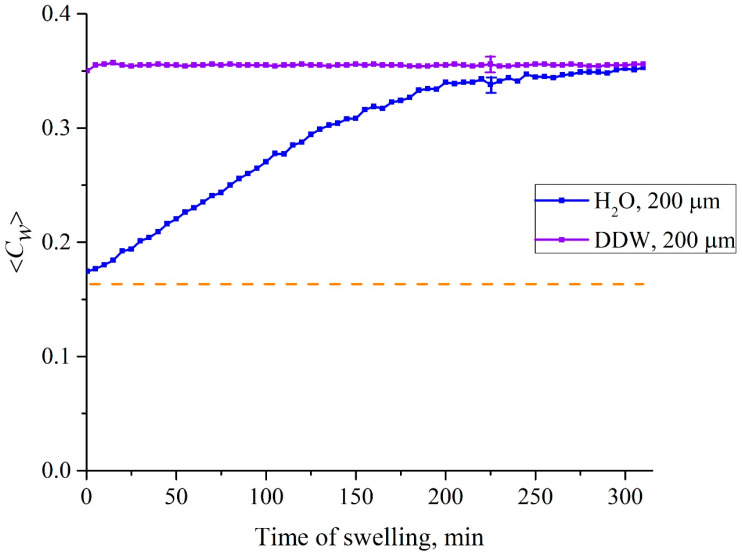
Concentration of water 〈*C_w_*(*t*)〉 vs. swelling time *t*, averaged over the distance *L* = 200 μm between the windows of the cell. The graphs are for DI water (blue curve) and DDW (magenta curve). The horizontal dashed line is related to the concentration water for dry Nafion (*C_w_*)_0_ = 0.174 (baseline).

**Figure 21 polymers-15-02214-f021:**
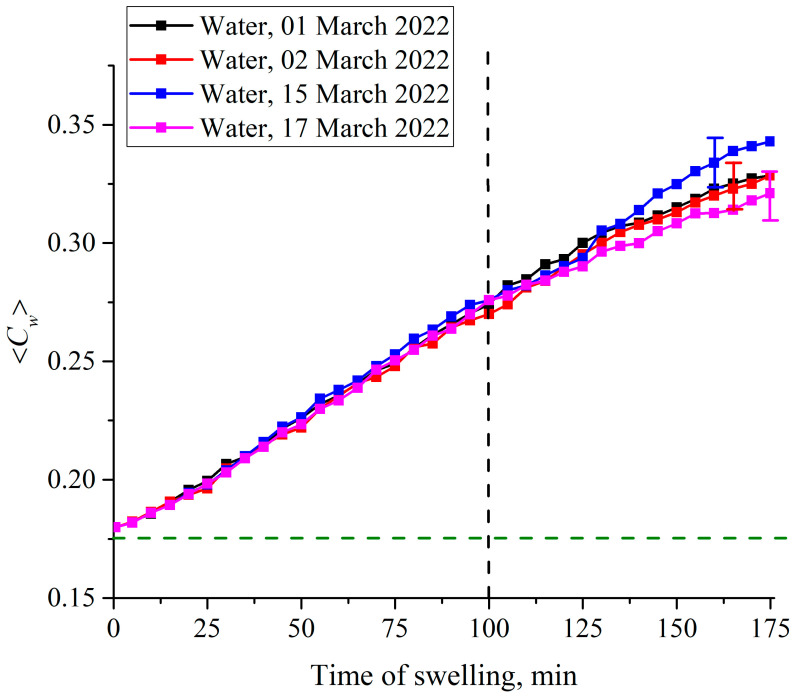
The concentration of water 〈*C_w_*(*t*)〉, averaged over the distance *L* between the windows of the cell, as a function of the swelling time *t* for DI water. The samples of this water were obtained on different days from various Milli-Q devices. The horizontal dashed line is related to the concentration of water for dry Nafion (*C_w_*)_0_ = 0.174 (baseline).

**Figure 22 polymers-15-02214-f022:**
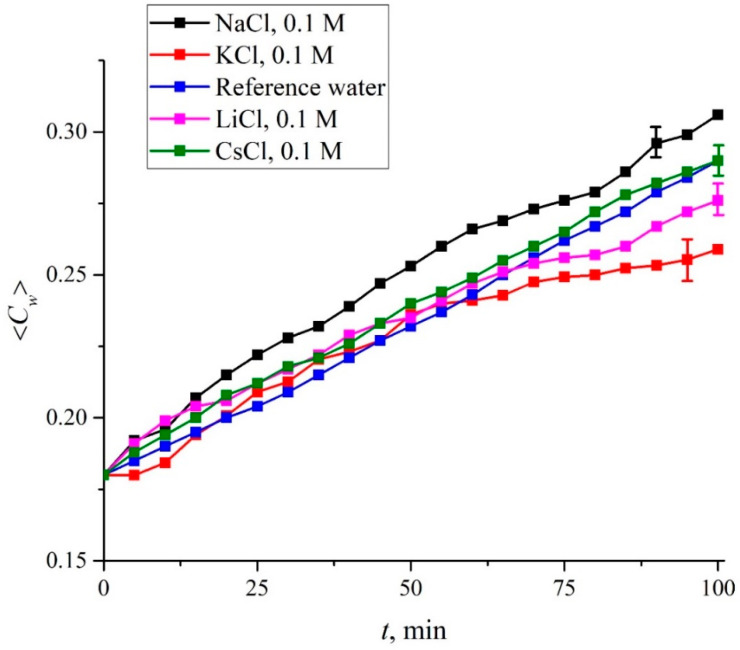
Dependence of the average concentration 〈*C_w_*(*t*)〉 for LiCl, NaCl, KCl and CsCl solutions; the ionic content is equal to 0.1 M. The reference dependence 〈*C_w_*(*t*)〉 for DI water is also shown.

**Figure 23 polymers-15-02214-f023:**
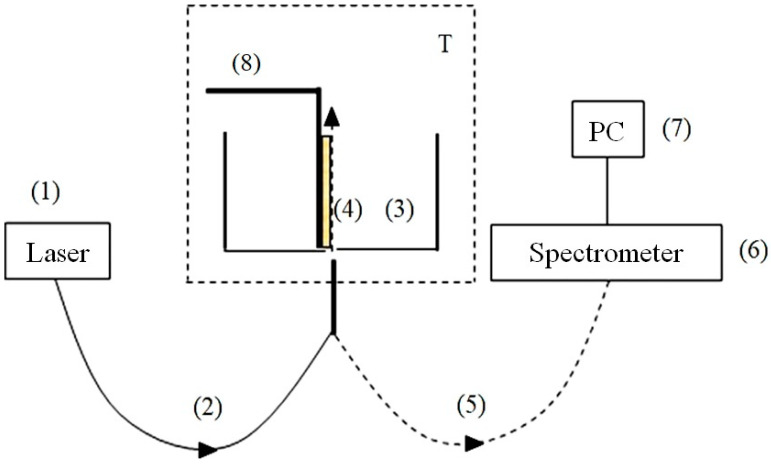
Schematic of the experimental setup for laser photoluminescence spectroscopy; second protocol. (1)—Laser for optical pumping; (2)—multimode optical fiber for transferring the pump radiation; (3)—Teflon cell for liquid samples; (4)—Nafion plate; (5)—multimode optical fiber for transferring the luminescence radiation; (6)—mini-spectrometer; (7)—computer; (8)—stepper motor.

**Figure 24 polymers-15-02214-f024:**
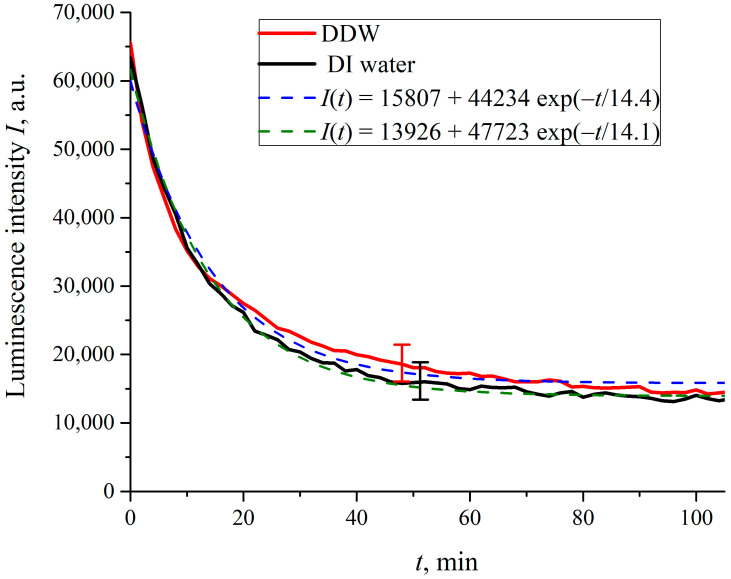
Intensity of luminescence *I*(*t*) vs. the time *t* of soaking in DI water and in DDW.

**Figure 25 polymers-15-02214-f025:**
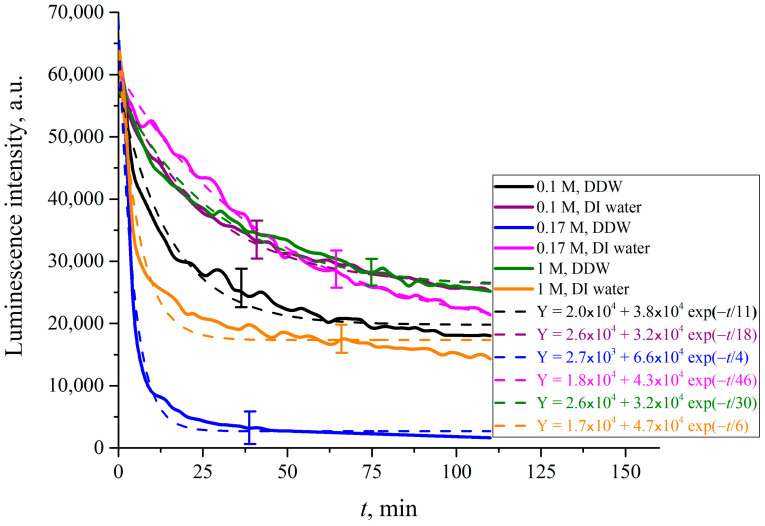
Luminescence intensity *I*(*t*) vs. the time *t* of soaking in the solutions of LiCl, based on DI water and DDW.

**Figure 26 polymers-15-02214-f026:**
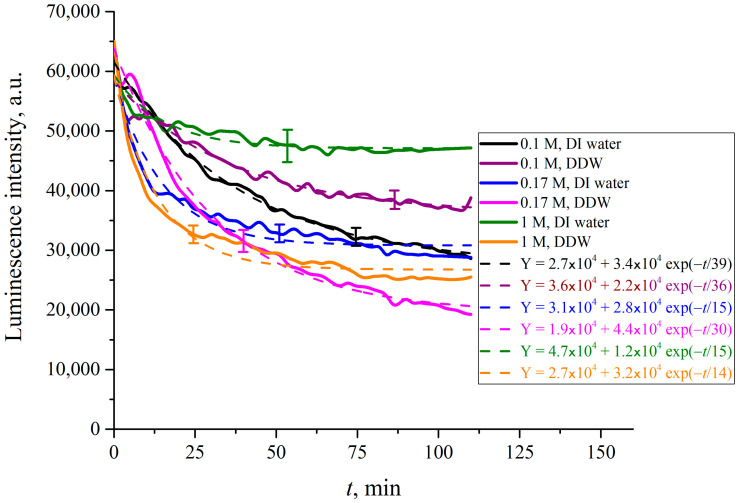
Luminescence intensity *I*(*t*) vs. the time *t* of soaking in the solutions of NaCl, based on DI water and DDW.

**Figure 27 polymers-15-02214-f027:**
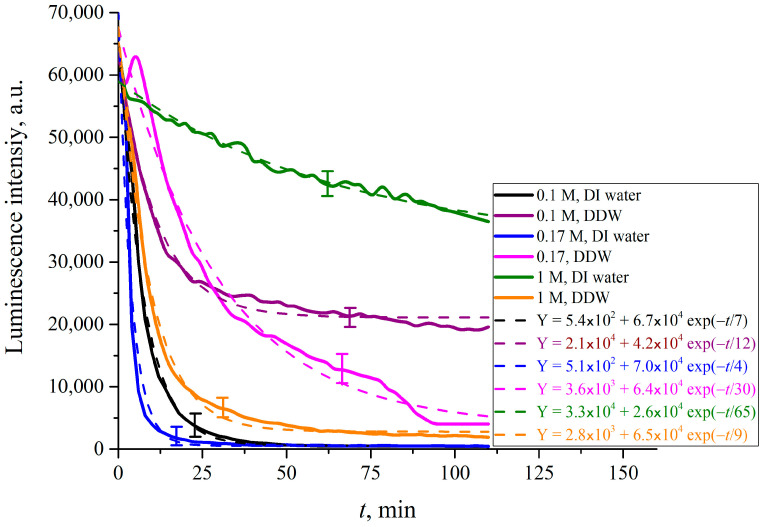
Luminescence intensity *I*(*t*) vs. the time *t* of soaking in the solutions of KCl, based on DI water and DDW.

**Figure 28 polymers-15-02214-f028:**
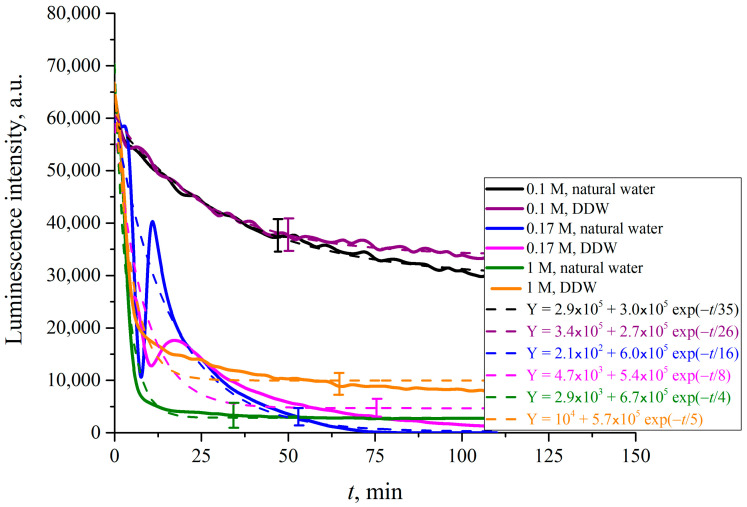
Luminescence intensity *I*(*t*) vs. the time *t* of soaking in the solutions of CsCl, based on DI water and DDW.

**Figure 29 polymers-15-02214-f029:**
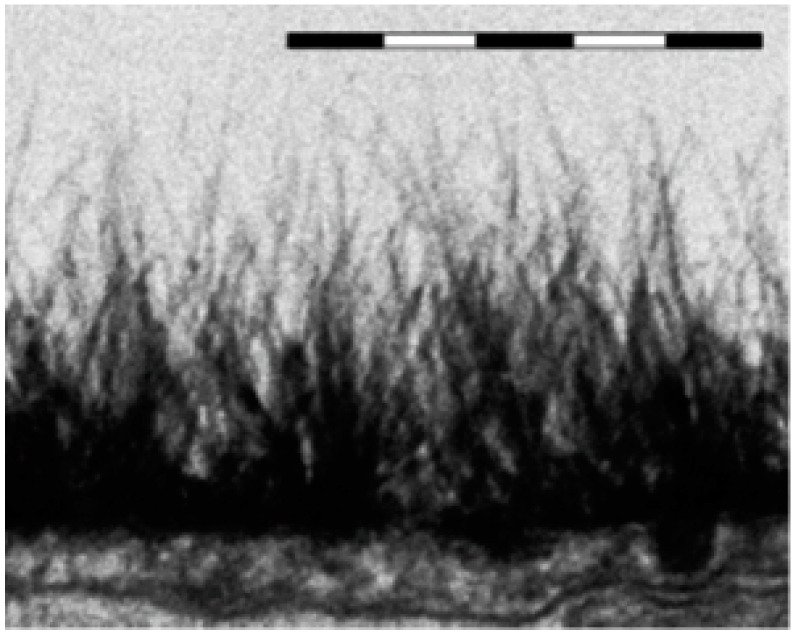
Electron microscopy pattern of the endothelial glycocalyx in rat myocardial capillary (bar = 0.5 μm). Reprinted from Figure 1c, Ref. [62] with permission from H. Vink et al., Copyright 2003, AHA Journals.

**Table 1 polymers-15-02214-t001:** *C* = 0.1 M.

	(Y_0_)_DI___water_ × 10^4^	(Y_0_)_DDW_ × 10^4^	(Y_0_)_DI___water_/(Y_0_)_DDW_
LiCl	2.6	2	1.3 ≈ 1
NaCl	2.7	3.6	0.3
KCl	5.4	2.1	2.6 ≈ 3
CsCl	2.9	3.4	0.9 ≈ 1

**Table 2 polymers-15-02214-t002:** *C* = 0.17 M.

	(Y_0_)_DI___water_ × 10^4^	(Y_0_)_DDW_ × 10^4^	(Y_0_)_DI___water_/(Y_0_)_DDW_
LiCl	1.8	2.7	0.7 ≈ 1
NaCl	3.1	1.9	1.6 ≈ 2
KCl	5.1	3.6	1.4
CsCl	2.1	4.7	0.4

**Table 3 polymers-15-02214-t003:** *C* = 1 M.

	(Y_0_)_DI___water_ × 10^4^	(Y_0_)_DDW_ × 10^4^	(Y_0_)_DI___water_/(Y_0_)_DDW_
LiCl	1.7	2.6	0.7 ≈ 1
NaCl	4.7	2.7	1.7 ≈ 2
KCl	3.3	2.8	1.2 ≈ 1
CsCl	2.9	1	2.9 ≈ 3

## Data Availability

The data presented in this study are available on request from the corresponding author.

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
