# Peer review of "Nafion: New and Old Insights into Structure and Function"

_polymers, 2023, doi:10.3390/polym15092214_

Round 1

Reviewer 1 Report

In this work, the authors have investigated the dynamics of swelling of the polymer ion-exchange membrane of Nafion in water with different deuterium content obtained in experiments with to photoluminescent and Fourier IR spectroscopy. This work should significantly contribute to the understanding of commercial Nafion CEM. I recommend considering its publication in Polymer without further revision.

Author Response

The Reviewer is thanked for the constructive feedback. We have improved the text in accordance with the Reviewer’ remarks. Our responds to the Reviewer’ criticism are highlighted by italic font.

Major comments

  1. While it is a review paper, the abstract needs to be shorter; please be more concise. Also, please consider checking the latest template (2023).

The Abstract has been significantly shortened in accordance with the latest template (200 words).

  1. The authors have focused on [1], which is a 2004 review paper. Can you consider mentioning more recent review papers as well? I suggest looking at a 2017 review paper from the same journal (https://pubs.acs.org/doi/full/10.1021/acs.chemrev.6b00159).

Thank you for this remark. In the revised version we mention the latest review: Kusoglu, A.; Weber, A.Z. New Insights into Perfluorinated Sulfonic-Acid Ionomers. Chem. Rev. 2017, 117, 987−1104 (Ref. [1] in the new version).

  1. (pg. 8-2nd paragraph) It would be helpful to mention why heparin and chondroitin were selected (e.g., bio-mimicking, etc.) instead of containing sulfonates.

The idea of investigating heparin and chondroitin sulfate is that these substances contain sulfonic group SO_3H. The additional reason of why we investigated the luminescence spectra of aqueous solutions of heparin and chondroitin sulfate is that heparin and chondroitin sulfate are well known drugs. Since we studied the properties of Nafion in the context of possible biological applications, it was interesting for us to compare the luminescence spectra of Nafion, heparin and chondroitin sulfate. The appropriate comment has been added to the text.

  1. Is the data for “2.3. Aspects…” a new data never been published?

These data were partially published in our recent study [32]. We have written a review of our previous works and the works of other authors. The articles to which we refer in our review have been widely cited.

  1. (pg 24) have you tried a different cell, distance, and conditions (temperature)? It seems hasty to assert it is due to the unwinding of the polymer strands.

Indeed, we have studied cells with a distance L between windows in the range of 180 – 220 microns. The experiments were carried out at room temperature, since in the intervals between particular measurements it was necessary to cool the samples, which were heated due to absorption. This review includes the results obtained for L = 200 microns. In addition, we studied cells whose windows had different degrees of roughness. For rougher cell windows, the time of the cavity collapse was significantly higher than for windows with high quality of polishing. These results are described in our recent papers [44, 45]. We did not include these results in the review so as not to overload the text. We have added an appropriate commentary to the new version of the manuscript.

  1. The summary section (pg 41) could be easier to read. I do not see the purpose of having multiple single sentences instead of full paragraphs.

This section has been completely rewritten and significantly shortened.

Minor comments

  1. Can you write out FTIR once and use FTIR instead of using Fourier IR? Also, stay with abbreviations once defined. For instance, the exclusion zone (EZ) and stay with EZ instead of rewriting the exclusion zone.

This was fixed throughout the text of the new version.

  1. Is there a reason why the chemical structure of Nafion is not a figure? Also, drawing the chemical structure might be better for readers.

The publication of other author's drawings, obviously, requires the permission of the copyright holders. It will take some time to obtain such permission. But we have to submit the revised manuscript within a limited time. We will not be able to obtain such permission by the date of submission of the new version.

  1. Multiple spacing issues and spelling issues. Please consider running with a checker.

Thank you for this remark. The spacing and spelling issues have been fixed throughout the text.

  1. Terminologies used for different types of water are confusing. For instance, this reviewer thinks ‘natural’ water should be ‘deionized water or DI water’, regarding H2O, D2O, and DDW terms as in pg 11. It makes more sense that DDW should be H2O because it should be pure H2O. And H2O should be just DI water, which consists of some D2O.

Natural water has been replaced by DI water throughout the text. At the same time, DDW is a stable term, so we decided to leave this term in the new version.

Reviewer 2 Report

The authors of “Nafion: New and Old Insights into Structure and Function” summarized their findings on the effects of deuterium on the unwinding of Nafion 117. While their findings are interesting, this reviewer believes they can be improved before publication.

Major comments

1. While it is a review paper, the abstract needs to be shorter; please be more concise. Also, please consider checking the latest template (2023).

2. The authors have focused on [1], which is a 2004 review paper. Can you consider mentioning more recent review papers as well? I suggest looking at a 2017 review paper from the same journal (https://pubs.acs.org/doi/full/10.1021/acs.chemrev.6b00159).

3. (pg. 8-2nd paragraph) It would be helpful to mention why heparin and chondroitin were selected (e.g., bio-mimicking, etc.) instead of containing sulfonates.

4. Is the data for “2.3. Aspects…” a new data never been published?

5. (pg 24) have you tried a different cell, distance, and conditions (temperature)? It seems hasty to assert it is due to the unwinding of the polymer strands.

6. The summary section (pg 41) could be easier to read. I do not see the purpose of having multiple single sentences instead of full paragraphs.

Minor comments

1. Can you write out FTIR once and use FTIR instead of using Fourier IR? Also, stay with abbreviations once defined. For instance, the exclusion zone (EZ) and stay with EZ instead of rewriting the exclusion zone.

2. Is there a reason why the chemical structure of Nafion is not a figure? Also, drawing the chemical structure might be better for readers.

3. Multiple spacing issues and spelling issues. Please consider running with a checker.

4. Terminologies used for different types of water are confusing. For instance, this reviewer thinks ‘natural’ water should be ‘deionized water or DI water’, regarding H2O, D2O, and DDW terms as in pg 11. It makes more sense that DDW should be H2O because it should be pure H2O. And H2O should be just DI water, which consists of some D2O

Author Response

(The authors gave the same response as above.)
